# HOLSEA-NL: Holocene water level and sea-level indicator dataset for the Netherlands

Kim de Wit[1], Kim M. Cohen[1], Roderik S. W. van de Wal[1,2]

[1]Department of Physical Geography, Utrecht University, P.O. Box 80.115, 3508TC Utrecht, the Netherlands

[2]Institute for Marine and Atmospheric research Utrecht, Utrecht University, 3584 CC Utrecht, the Netherlands

*Correspondence to*: Kim de Wit (k.dewit@uu.nl)

**Abstract.** Deltas and coastal plains worldwide developed under the influence of relative sea level rise (RSLR) during the Holocene. In the Netherlands, Holocene RSLR results from both regional sea-level rise and regional subsidence patterns, mainly caused by glacial isostatic adjustment (GIA: Scandinavian forebulge collapse) and longer-term North Sea Basin

tectono-sedimentary subsidence. Past coastal and inland water levels are preserved in geological indicators marking the gradual drowning of an area, for example, basal peats. Such geological water-level indicators have been used in the Netherlands for varying types of research. However, uniform overviews of these data exist only for smaller local subsets rather than for the entire Netherlands. In this paper, we present a data set of 712 Holocene water-level indicators from the Dutch coastal plain that are relevant for studying RSLR and regional subsidence, compiled in HOLSEA workbook format (De Wit and Cohen,

2024). This format was expanded to allow for registering basal-peat type geological indicators, documenting Dutch-setting specific parameters and accompanying uncertainties, assessing indicative meaning, and appropriately correcting the raw vertical positions of the indicators. Overall, our new, internally consistent, expanded documentation provided for the water-level indicators encourages users to choose the information relevant to their research and report RSLR uncertainties transparently. From the indicators, 59% was collected in 1950-2000, mainly in academic studies and survey mapping

campaigns; 37% was collected in 2000-2020 in academic studies and archaeological surveying projects; 4% was newly collected (this study), the latter mainly in previously under-sampled central and northern Netherlands regions. Prominent regional differences exist in the vertical position and abundance of the indicators. Older indicators in our data set are primarily located in the deeper seaward area of the Netherlands. These indicators correspond well with previous transgression reconstructions partly based on the same data. The younger, landwards set of indicators in the Rhine-Meuse delta inland and

Flevoland regions corresponds with the transgression phase reaching further inland, from 8000 cal. BP onwards. Northern indicators of Middle Holocene age (8-5 ka cal. BP) generally lie 2-3 meters lower than those in the south. This difference is less for younger data, showing spatial and temporal variation in RSLR throughout the Netherlands.

## 1 Introduction

Holocene water-level indicators have been the subject of research in the Netherlands for decades. Previous studies collected geological water-level indicators for relative sea-level reconstructions (Jelgersma, 1961; Van de Plassche, 1982; Hijma and Cohen, 2019), geological mapping of the Dutch coastal-deltaic plain (e.g. Berendsen and Stouthamer, 2002), wetland palaeoenvironmental reconstructions (e.g. Vos, 2015), and archaeological excavation and dating (e.g. Verbruggen, 1992). These activities resulted in the accumulation of an extensive amount of primary water level indicator data (e.g. Jelgersma, 1961; Van de Plassche, 1982; Meijles et al., 2018; Hijma and Cohen, 2019; Quik et al., 2022). Geological water-level indicators also carry information for studying various types of subsidence, namely their depth positions with increasing age, as well as the local and regional variabilities therein (Kiden et al., 2002; Cohen, 2005; Van Asselen, 2011; Koster et al., 2017). The total reconstructed relative water-level rise signal can be separated into a Holocene water level rise and a land subsidence history. This is possible by evaluating geological records in combination with independent sea-level and subsidence reconstructions, and geophysical modelling simulation output. As such, subsets of water-level indicators are used to verify location-specific RSLR output of glacialsostatic adjustment (GIA) modelling, which incorporates ice sheet deglaciation history and Earth-rheology models to resolve RSLR globally (Lambeck, 1995; Kiden et al., 2002; Shennan and Horton, 2002; Vink et al., 2007; Bradley et al., 2011).

The abundance of geological palaeo-water level observations in the Netherlands creates a unique opportunity to study Holocene differential subsidence across the entire coastal plain. Currently, an integrated overview of the geological indicator data with consistent documentation is lacking, mainly due to the above-mentioned diversity in data usage purposes. To bridge this gap, we created a systematic overview of the current vertical position of geological indicators; applied a uniform set of consecutive vertical corrections, such as for water depth and compaction; and constructed a consistent error propagation workflow. The aim of this paper is to present this newly compiled dataset of water-level indicator points, serving the study of regional relative sea- and groundwater-level rise in the Netherlands over the Holocene and generically disclosing and arranging this rich data for inclusion in European and global scale Holocene RSLR and coastal plain accommodation studies.

Building on previous work, the geological data compilation includes a focused review of the usability of legacy data for relative water-level reconstructions. The documentation follows the HOLSEA workbook format for Holocene relative sea-level data (Hijma et al., 2015; Khan et al., 2019). The HOLSEA workbook is a versatile data reporting format that includes correction specifications and metadata (e.g. Hijma and Cohen, 2019; Bungenstock et al., 2021; Creel et al., 2022). It categorises relative sea-level data entries in 'sea-level index points' (SLIPs), sea-level position bounding 'upper limiting data' (ULD) and 'lower limiting data' (LLD).

From our study area, a subset of 104 basal peat dates from the Rhine's lower delta was previously compiled and published in HOLSEA format (Hijma and Cohen, 2019): their 'Rotterdam' data set). Hundreds of similar dates containing potential SLIP, ULD and LLD information, which were not yet compiled, assessed, and disclosed in HOLSEA format, exist in publications from the 1950s-2010s, institutional databases, contextual reports and unpublished data. To fill this gap, this paper expands the HOLSEA-format covered data from the Netherlands to a total of 712 samples, further referred to as the HOLSEA-NL data set.


While compiling the HOLSEA-NL data set, attention was given to enhancing data usability. First, various vertical correction components are specified that users are recommended to apply, such as fen/swamp water depths and peat decompaction. Additional correction components, such as palaeo-tidal range and long-term background land motion, are optional and can be applied depending on the specific application. Next, the indicative meaning of the water-level indicators was reviewed to

assess which samples qualify as SLIPs, ULD and LLD. Of these, the ULD category was expanded to allow for characterising various groundwater level data points, which are relevant for reconstructing delta plains and inner lagoon peat land fringe regions as well as identifying regional trends of subsidence (e.g. Cohen, 2005). The indicative meaning of water-level indicators differs per type of deposit and past geographical setting. It is determined based on the sedimentary and biotic facies, the succession criteria on individual geological sampling locations, the spatial position of the sample and criteria on ensembles

of samples (e.g., outlier analysis, seaward locations priority over inland locations). Lastly, the depth of a past groundwater level (GWL) is calculated from the sample depth and offsetting this based on the sample indicative meaning. For SLIPs and tidally linked ULD and LLD, the past GWL can be upgraded to a past mean sea level (MSL), which is referred to as a relative sea level (RSL) relative to present-day MSL. To account for the different reference water levels (RWL), we documented both the GWL and the RSL in our data set.


The paper proceeds further as follows: Section 2 provides an overview of the study area and its geological setting. In sections 3 to 6, the set-up of the data set is described. Section 3 describes the data inventory, including data requirements and a description of different indicator types. Details on the age-depth positions, the systematic vertical corrections and additional optional adjustments are covered in Sect. 4. Section 5 presents an overview of the data, with regional and categorical

breakdowns. The last section discusses potential applications and limitations of the data.

## 2. Study area and geological setting

The Netherlands is located in the southern part of the North Sea Basin. During the Holocene, this area was strongly influenced by RSLR resulting from the deglaciation of land ice and regional subsidence caused by sinking of the North Sea sedimentary basin (Kooi et al., 1998) and glacial isostatic adjustment (GIA) remaining from the last glacial period (Lambeck, 1995; Kiden

et al., 2002; Vink et al., 2007; Bradley et al., 2011). The relatively shallow depth of the southern North Sea and the variation in sediment fluxes throughout the Holocene played an important role in the development of the coastal area of the Netherlands.

This setting and sedimentation history also determined how and when different water level indicators could form and, therefore provides relevant context for understanding the variability in indicative meaning of geological water-level indicators (Sect. 3-5). Whenever a peat layer formed on top of consolidated sediments in favourable landscape conditions (see Sect. 3) and was

preserved and sampled (Sect. 1), it is suitable as a water level rise indicator (SLIP, ULD primary (= tidal), ULD fluvial, ULD local inland in Fig. 2), and in some cases even as sea-level indicator (LLD, SLIP, ULD primary in Fig. 2). The geological development of the Netherlands is based on hundreds of thousands borehole observations, thousands of radiocarbon dates, further dating, and paleoenvironmental evidence, collected in parallel regional and national campaigns by multiple surveying agencies (Pons and Wiggers, 1960; Zagwijn, 1986; Berendsen and Stouthamer, 2001; Van der Meulen et al., 2013; Vos, 2015b;

Cohen et al., 2017a, b; Pierik and Cohen, 2020). An important aspect of these studies was to provide insight into the timing and the rates of water level rise by dating coastal peats (basal and intercalated; transgressive and regressive). As a result, a detailed reconstruction of peat formation and further Holocene development in the Dutch coastal plain is possible.

At the onset of the Holocene (11,650 cal. BP), the water level in the North Sea was still low, with the shallow southern seafloor

largely exposed. Pleistocene depositional landforms constituted the Netherlands in the form of periglacial aeolian dune fields, coversands, and river valleys dissecting older terraced plateaus and hills (Figure 1a; Figure 2a). Below the Holocene coastal plain of the western Netherlands, two East-West running Late Pleistocene valleys (palaeovalleys) are of relevance (Busschers et al., 2007; Vos, 2015b; Peeters et al., 2016; Koster et al., 2017)). As they are the lowest elevated areas, they were the first to be affected by marine transgression and hence they developed the thickest Holocene records: The southern one hosted the

Rhine-Meuse system of the time, which was joined by Scheldt in the near offshore. The northern one was drained by the Overijsselse Vecht underfit system (Figure 1a). Locally, inland dunes formed along river channels of each of these systems (Bennema and Pons, 1952; Wiggers, 1955; Gotjé, 1993; Berendsen and Stouthamer, 2001; Wolfert and Maas, 2007; Kasse and Aalbersberg, 2019), features that Holocene water-level rise studies have specifically targeted (see below). Below the northern coastal plain, smaller palaeovalley systems of the Boorne and Hunze and Ems system are featured (e.g. Vos, 2015b;

Meijles et al., 2018). Interfluve relief in the centre and north of the country shows push-complexes and a main till-sheet that are remnants of glaciation and deglaciation in the Saalian (the penultimate ice age; ca. 150,000 years ago, within MIS6). The latter till sheet, called the 'Drenthe plateau' (Figure 1a), forms a shallow aquitard, affecting pre-transgression groundwater tables (Van den Berg and Beets, 1987; Quik et al., 2021). Present-day relief expression of the till sheet in the North, the central ice-pushed ridge complex and southwest Pleistocene fluvial terraces, determined the inland boundary of the area of interest in

this paper (Figure 1b).

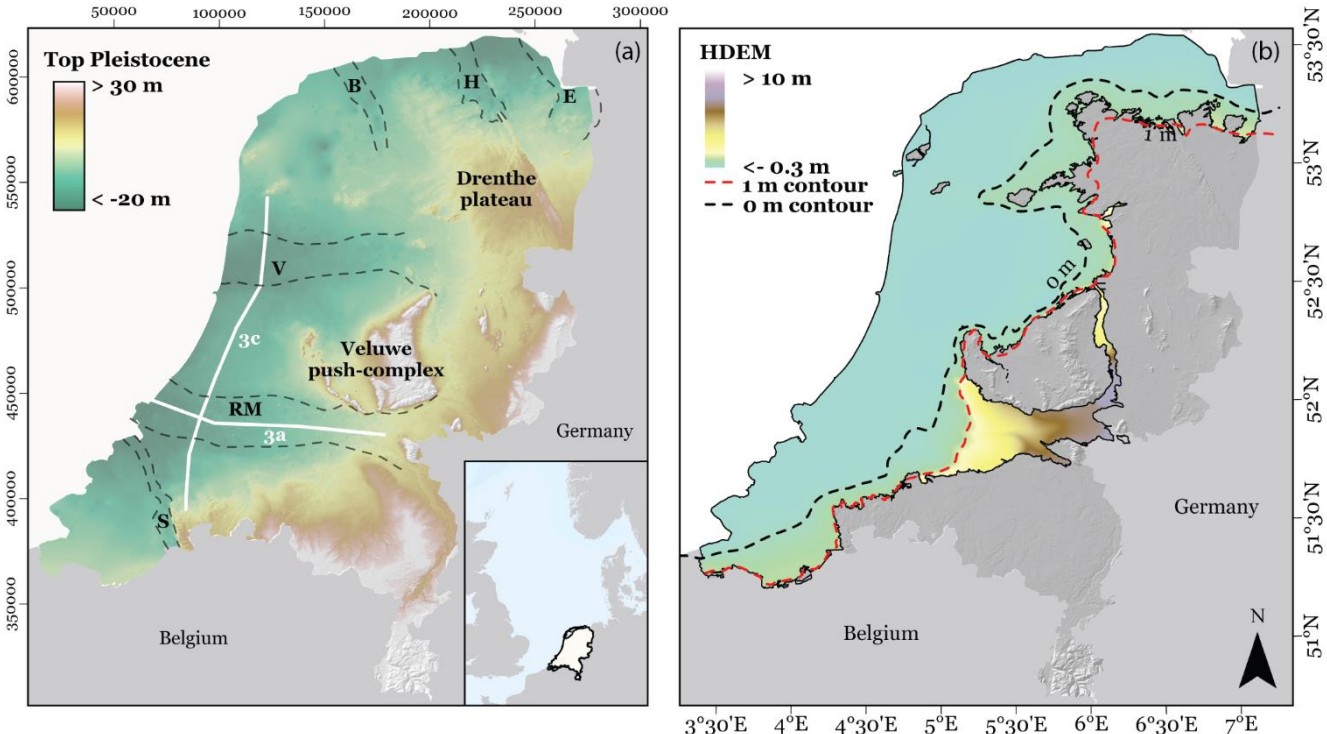

**Figure 1 (a)** Digital elevation model of the top of Pleistocene deposits, corrected for Holocene erosion (Stafleu et al., 2012; Cohen et al., 2017b, a; Koster et al., 2017). The labels and dashed outlines show the location of the main palaeovalleys, B: Boorne, H: Hunze, E: Ems, V: Vecht, RM: Rhine-Meuse, S: Scheldt; The white lines show the orientation of the example cross-sections from Fig. 3; **(b)** High stand digital elevation model (HDEM), regional groundwater table surface reconstructed for 1000 cal. yr. BP (Cohen, 2005; Cohen et al., 2017a, b; locally modified). The 0 m MSL contour (black dashed) and +1 m MSL contour (red dashed) are shown. The study area contour is adopted from Pleistocene landform outlining in national geomorphological and archaeological landscape mapping (as in Cohen et al., 2017a, b) and outside the Rhine-Meuse delta roughly coincides with the +1 m elevation contour line. Projection: RD (EPSG 28992)

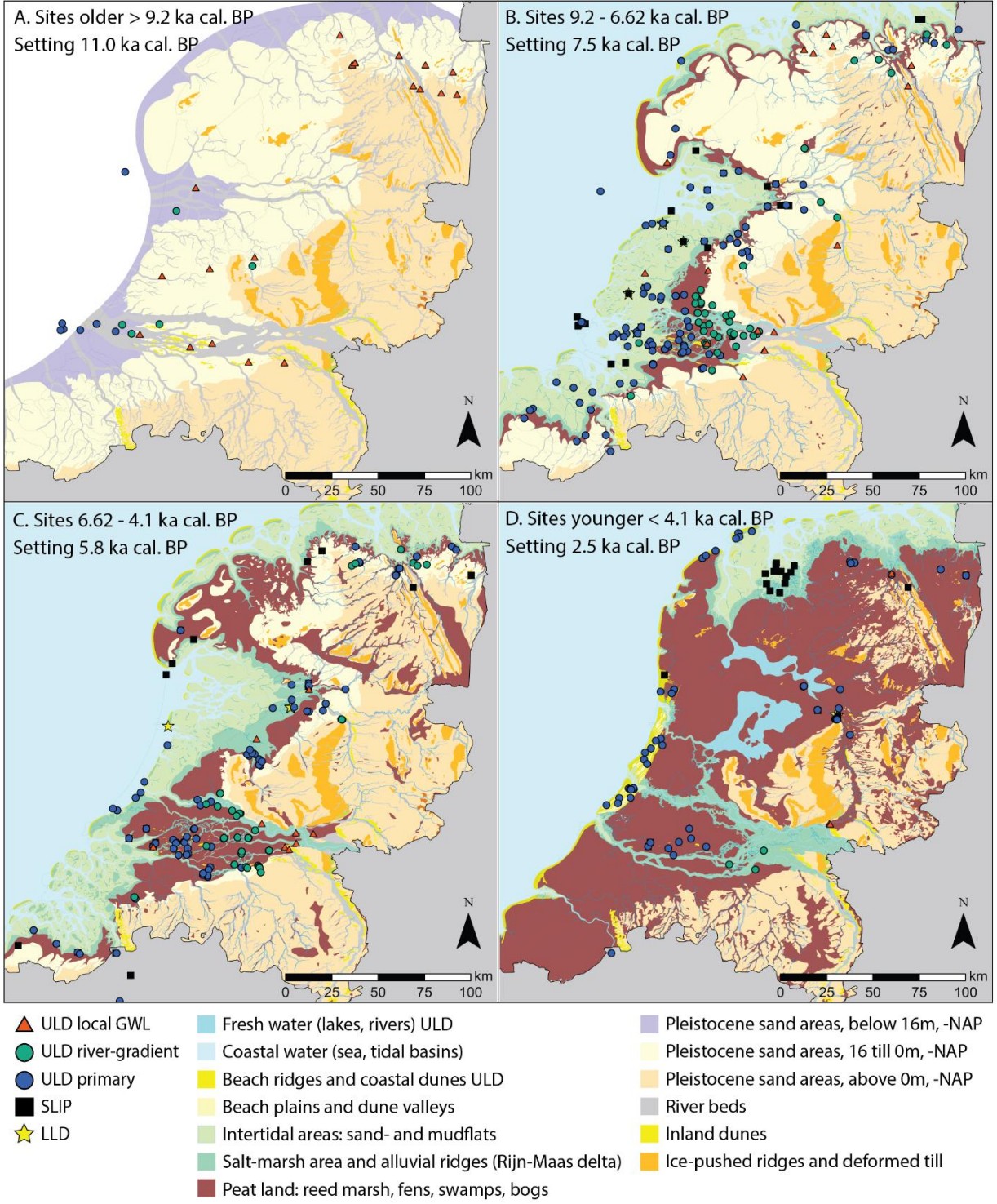

A. Sites older > 9.2 ka cal. BP
Setting 11.0 ka cal. BP

B. Sites 9.2 - 6.62 ka cal. BP
Setting 7.5 ka cal. BP

C. Sites 6.62 - 4.1 ka cal. BP
Setting 5.8 ka cal. BP

D. Sites younger < 4.1 ka cal. BP
Setting 2.5 ka cal. BP

Legend:
- ULD local GWL
- ULD river-gradient
- ULD primary
- SLIP
- LLD
- Fresh water (lakes, rivers) ULD
- Coastal water (sea, tidal basins)
- Beach ridges and coastal dunes ULD
- Beach plains and dune valleys
- Intertidal areas: sand- and mudflats
- Salt-marsh area and alluvial ridges (Rijn-Maas delta)
- Peat land: reed marsh, fens, swamps, bogs
- Pleistocene sand areas, below 16m, -NAP
- Pleistocene sand areas, 16 till 0m, -NAP
- Pleistocene sand areas, above 0m, -NAP
- River beds
- Inland dunes
- Ice-pushed ridges and deformed till


**Figure 2 Geological water-level indicator site locations(this paper and data set; split into four time slices) plotted over a national palaeogeographical map series (Vos et al., 2020). The indicators shown have an age surrounding the date stamp of each selected map.**

During the Early Holocene (11.6 – 8.3 ka cal. BP), the sea level increased from 60-50 m to approximately 25-20 m below the current MSL, which steadily inundated the Southern North Sea, eventually establishing coastlines in the vicinity of the modern one. River waters and inland flood basin groundwater tables were affected by the downstream rise, resulting in a decrease of the river gradient and concurrent rise of the inland groundwater levels. This initiated the growth of peat in swamps and fens

on former floodplains and along the drowning valley edges. From dating such transgressive peats ('basal peats'), overlain by later marine deposition, earliest water-level indicators are obtained in the near offshore (Figure 2a), and eventually across the study area, following the palaeovalleys as gateways for the transgressive peats (Figure 2b).

As the Middle Holocene (8.3 – 4.2 ka cal. BP) commenced, the decelerating sea-level rise pushed the transgression further

landwards, as is evident from basal peats buried by tidal muds. Simultaneously, a beach barrier system and back-barrier Rhine-Meuse delta established (Figure 2b,c) (Berendsen and Stouthamer, 2002; Vos, 2015b). The former valleys became tidal-dominated embayments, with intertidal shoals and fringing supratidal marshlands on the landward side (Figure 2b,c). This changed when the beach ridge complex matured during the Middle Holocene. The decreasing rate of sea-level rise and the supply of sediments caused most inlets of the Western Netherlands coastline to close around 5.8 ka cal. BP (Figure 2c) (Beets

and Spek, 2000; Hijma & Cohen, 2011). These developments and the water supply from the Rhine-Meuse delta helped turn the back-barrier area into a freshening lagoon in which widespread peat formation initiated. Areas just north remained tidally dominated for some 1500 years longer. The flood basin of the so-called Bergen tidal inlet transformed into a lagoon, gradually developing an expanding freshwater peat rim (Figure 2c) as the inlet decreased in size from 5 ka cal. BP onwards (Beets and Van der Spek, 2000). Around 3.5 ka cal. BP, also the tidal inlet of Bergen finally closed, promoting the further spread of peat

land in the coastal plain (Figure 2d). In the very north, a series of tidal inlets remained functional because of the relatively stronger subsidence and relative lack of sediment supply in this area. Such features  still persist today as the several inlets separating the Frisian Islands. The adjacent tidal basins form the Wadden Sea, its supratidal salt marshes and coastal peatlands currently bordering the mainland of the Northern Netherlands (Figure 2d). Although peat formation continued, it is difficult to find young peat layers (< 2.5 ka ca. BP). This is because many of the young peat layers disappeared, partly due to erosion by

rivers or the sea, but predominantly due to large-scale peat excavations that occurred since the Middle Ages when peat was mined extensively (Pierik et al., 2017). Human activity also caused large-scale soft soil subsidence. This is still ongoing, mainly caused by the lowering of the groundwater level, resulting in compaction of clay and peat, and peat oxidation (Van Asselen, 2011; Erkens et al., 2016). To minimise the influence of natural and human induced soft soil subsidence on the elevation of the water level indicators for GWL and RSL study, mainly basal samples were considered (see next Sections).


In summary, during the Holocene peat formed in abundance throughout most of our study are, at basal positions in the Holocene wedge and at shallower positions. From 9.2 till 5 ka cal. BP under transgressive influences, lasting longest in the North and from 5.5 ka cal. BP onwards under regressive circumstances, starting from the south and moving northwards. Much of the peat from the younger period disappeared, mainly due to human activity in the past 1000 years, leaving a gap in the record for the most recent period.

### 3. Data inventory and intake

The water-level indicators collected in the Netherlands for different purposes over the past decades (Sect. 1) are of different types, but the great majority (640 out of 712 entries in the database) consist of sampled and dated basal peat layers from a range of settings (Figure 2 and Figure 3). In coastal areas, the dated contact of basal peat layers burying older substrate has long been used as an indicator in sea-level research, notably where basal peat could be collected along flanks of inland dunes (Sect. 2) (e.g. Jelgersma, 1961; Van de Plassche, 1982; Hijma and Cohen, 2019). Just as well, dating of the contact of the basal peat top with transgressive muds has been used as such (Hijma and Cohen, 2010, 2019). In archaeological studies, the ages of basal peat layers helped to constrain the ages of archaeological findings, again notably on inland dunes (Verbruggen, 1992). In addition, basal and intercalated peat layers have been collected to determine the timing of branch avulsions in the Rhine delta and compare channel sedimentation levels with contemporary flood basin water levels (Stouthamer and Berendsen, 2000; Berendsen and Stouthamer, 2001). Intercomparison of dates from basal and intercalated positions has been used for studying auto-compaction in unconsolidated sediments (Van de Plassche, 1980; Van Asselen et al., 2009; Van Asselen, 2011). This section describes the requirements applied to select the legacy data to ensure a uniform input relevant for relative water-level reconstruction research. The second part of this section describes the main types of water level indicators included in the data set.

### 3.1 Data requirements

Basal peat beds and contacts have been the preferred geological water-level indicators because of the combination of their radiocarbon dating potential (see below) and the fact that the projected vertical position of the indicator is minimally influenced by post-depositional compaction processes, especially those resting on a palaeosurface in sandy Pleistocene substrate. This Pleistocene surface was exposed before peat formation set on, and had experienced initial compaction and pedogenic consolidation since deposition. Residual compaction of the Pleistocene deposits and those below (see Kooi et al., 1998) is considered a part of the separately specified tectono-sedimentary background subsidence component (see Sect. 4.1.3). Provided that the basal peat bed is up to a few decimeters thick, the minimal effect of post-depositional compaction applies to indicator levels sampled from the base, middle and top of the peat bed (e.g. Hijma and Cohen, 2019). Vertical decompaction correction and the associated uncertainty are smallest for base basal peat dates (decompaction uncertainty ~10 cm; Berendsen et al., 2007). Given the wealth of basal peat data available (research cited in Sect. 1), samples from intercalated peat layers from

shallower positions than the basal peats have been excluded here because of the larger decompaction uncertainty associated with these samples (for more peat data, see compilations such as Berendsen and Stouthamer, 2001). In some areas, relatively inland, peaty beds overlying the Pleistocene subsurface can reach a considerable thickness. From such peats, only levels sampled within 1 m from the Pleistocene substrate were included.

The inland boundary of the study area (Figure 1b) further constrains whether data are included. Studies sampling peat at inland locations above +1 m MSL have generally been excluded from the inventory, as peat formation there is often linked to locally perched groundwater levels and fluctuations therein (e.g. Hoek, 1997; Quik et al., 2021), particularly for areas in the north of the Netherlands with glacial till in the subsurface. Similarly, samples from pingos were excluded as well (for example those included in Quik et al., 2021). In contrast, the coastal plain water levels are graded to the regionally increasing water level, resulting in suitable regional water level indicators (Van de Plassche, 1995a). In the Rhine-Meuse river valley, samples up to the 10 m contour line have been included. Previous studies have shown that these types of samples show a regional trend in groundwater level as well, which can be linked to the sea-level and subsidence history and the evolution of the area from flood basins to tidal inlets, closed lagoons and large-scale peatlands (Van Dijk et al., 1991; Cohen, 2005).

The study area is further bounded towards the offshore (e.g. Figure 2a,b), where the inventory cut-off has been arbitrary. The dataset contains some basal peat sampled off the Holland coast and in the Wadden Sea – as did recent regional compilation studies for these sectors (resp. Meijles et al., 2018; Hijma and Cohen, 2019). Besides, the study area is bounded by national borders at critical locations in the southwest (Flanders, Belgium) and northeast (Niedersachsen, Germany). Where the Holocene fill of palaeovalleys extended across borders (Figure 2), legacy data was included to allow cross-verification with data from within the study area. This is the case for a few data points from the Scheldt (following Kiden et al., 2002) and Ems palaeovalleys (following Behre, 2007).

A last physical boundary is the upper limit where basal peat layers can be found, which is not a prescribed boundary but a consequence of the reclamation history of the study area (see Sect. 2): extensive peat excavation since 1000 years ago has degraded Late Holocene peats top down, which imposes a soft (i.e. spatially variable) upper temporal boundary making basal peat samples younger than 3500 years rare and younger than 2500 years very rare (Van de Plassche, 1982; Cohen, 2005). To overcome this limitation, some non-basal peat sea-level index points (13 dated shells sampled from below soles of raised mounts; Frisian terp archaeological sites) were included in the data set (Vos and Nieuwhof, 2021).

To conclude, the availability of metadata played an important role in including data in the database. An effort was made to trace any missing information, and when successful, the data points were added. Retracing age-depth data and meta-information included information on originally applied vertical offsets and corrections, to compare this with the uniformly applied and recalculated corrections in the HOLSEA workbook. Further information was retraced to specify and calculate the

uncertainty of different components for each data point. Points with insufficient information on location, depth and age determination, and associated uncertainties were left out.

## 3.2 Reported sampling methods and depth accuracy

The primary depth information in the dataset includes measurements of the surface elevation, the sample depth along the core and the thickness of the subsampled layer. This is stored in standard fields of the HOLSEA workbook, together with uncertainties and meta-information on the type of coring and type of surface elevation measurement (e.g., levelled to a benchmark, national LiDAR-datasets derived). The vertical datum to which depth is expressed is the Dutch national datum (NAP), which is approximately equal to 20[th] century MSL (e.g. Vermeersen et al., 2018).

Basal peats have been sampled using a variety of methods, such as hand coring, mechanical coring or excavation. The sampling method and elevation determination method affect the specification of the sample depth below surface elevation, and different uncertainties are associated with different methods. The uncertainties related to determining the absolute elevation of a core or section were assigned based on the acquisition method when not previously reported (e.g., levelling: ±0.02; and since the 21st century DGPS: ±0.01). The HOLSEA workbook provides a detailed breakdown of uncertainties related to sample

acquisition. Often, sampling (e.g., levelling and benchmark) uncertainty is combined in a single value, resulting in seemingly different uncertainty values. However, the total vertical uncertainties related to sample acquisition add up to similar values as previously reported.

The uncertainties for determining the depth of a sample in a core or section, are assigned separately. The overall error related

to measuring the sample depth in the core (sample-position accuracy) is set at 0.02 m following the estimated error found by Berendsen et al. (2007) when sampling from a core. For hand-cored samples, non-vertical drilling offsets are accounted for by adding an additional unidirectional uncertainty of -0.02 m per meter coring depth (Törnqvist et al., 2004; Hijma and Cohen, 2019), increasing only the upward component of the total vertical error. Depth uncertainty due to core shortening/stretching during sampling and initial storage is separately considered. This is set to ±0.05 m for hand and mechanically collected cores

alike, following Hijma and Cohen (2019).

In general, the depth uncertainty terms combined are smaller than the offsets and uncertainties depending on assigned indicative meaning (Sect. 3.3 and Sect. 4.2), decompaction (Sect. 4.1) and further vertical position corrections (Sect. 4.3 and 4.4). The exceptions are the samples taken offshore, for which the depth uncertainty terms are higher due to the additional

water depth uncertainty. For basal peat samples ultimately classified as SLIPs, the depth uncertainty terms account for about 20% of the total RSL depth uncertainty, while for basal peat samples ultimately classified as ULD, this accounts for about 25% of the GWL depth uncertainty.

### 3.3 Indicative meanings

Indicative meaning refers to the relation between the depth of deposition of the indicator and the water level at the time of deposition (Shennan, 1982; Van de Plassche, 1986; Hijma et al., 2015). Where it regards the indicative meaning of sampled basal peat, the type of peat collected (botanical composition, sedimentology, clastic admixture) as well as aspects of the geological setting (location in palaeolandscape and associated hydrological regime) contribute to specifying the indicative meaning of that sample. Bos et al. (2012) provided a classification key for organics and an overview of the distribution of basal peats underlying the Rhine-Meuse delta. They mapped the different facies (peat types) and distinguished between tidally (reed, clayey), river-flooding (woody), and seepage- or precipitation-dominated (fens, bogs) hydrological regimes in the downstream, inland, and rim sectors of the Rhine-Meuse palaeovalley, respectively. Van de Plassche (1982, 1986), Kiden (1995), Kiden et al. (2002), Makaske et al. (2003) and Van de Plassche et al. (2005, 2010) developed generic indicative meaning attribution schemes for Dutch settings, which were further developed for the Rhine-Meuse basal peats by Cohen (2005), Berendsen et al. (2007) and Hijma and Cohen (2019). In HOLSEA terms: any basal peat sample can be attributed a GWL-related indicative meaning forming an upper limit to MSL, a so-called ULD. In specific cases, these can be upgraded to a sea level related indicative meaning, allowing to define a SLIP.

### 3.3.1 GWL vs MSL indicative meaning

The diverse peat types that make up basal peat beds formed under different hydrological regimes with varying year-round water depths (e.g., woody swamps, reed marshes, sedge fens, mossy blanket bogs). For this reason, each sample is assigned an indicative meaning and indicative range according to Table 1 (Törnqvist et al., 1998; Makaske et al., 2003; Cohen, 2005; Berendsen et al., 2007; Bos et al., 2012; Hijma and Cohen, 2019). Based on that indicative meaning, the reference water level (RWL) is calculated, which represents the height of the water level at the time the water-level indicator was formed. The RWL is the midpoint of the indicative range (IR). This classification is the first step for relating the sample to a former GWL. A second step is determining if the GWL is linked to a marine-relatable RWL such as Mean High Water (MHW), e.g. for sites fringing a contemporary lagoon or estuary. If this is the case, the sample is also relatable to a past MSL (RSL) based on palaeo-tidal conditions (Sect. 4.1.2), and may define a SLIP.

The indicative water depth specifications of each peat type propagate into the eventual age-depth values as one of several vertical correction terms. The palaeo-water depth specification and uncertainty (Table 1) are based on the range of multiannual variation in the seasonally fluctuating water levels. For example, bog peats are ombrogenic, mossy, primarily rain-fed peat bodies, formed around a local water table (palaeo-water depth $= 0 \pm 0.1$ m) perched just above regional water levels. Fen-wood and fen peats are formed in varying hydrological settings: rain, river, and/or seepage-fed. Their palaeo-water depth corresponds to a regional water level, graded to inland past water levels from rivers and seepage zones and to lagoonal and deltaic flood basin water levels in the coastal zones. Fen-wood peats in the Netherlands are typically Alder wood dominated, though they

also contain moss, sedges, and reeds, reflecting the vegetation of former swamps, particularly common in river-flooded areas. In these environments, dead plant material accumulated on the peat's surface layer (the acrotelm), where the groundwater table remained at or near the surface for most of the year (palaeo-water depth = 0 m ± 0.1 m). Fen peats are often sedges and reed-dominated, with dead plant material accumulating just underwater and with an estimated acrotelm palaeo-water depth of 0.3 ± 0.2 m. This water depth varies depending on composition and site type, e.g. for "Fen peat on inland dune flanks", palaeo-water depth = 0 ± 0.2 m. For undetermined peat types, an intermediate estimated palaeo-water depth is assumed with a slightly larger uncertainty (0.2 ± 0.3 m). Table 1 also includes organic subaquatic accumulated, gyttjaic deposits, which are LLD and potential SLIP data points, when encountered topping basal peats (3 out of 32 gyttjaic samples in the dataset). For brackish mollusca and charcoal beds traced along a dune flank, the indicative meaning is determined separately per case in line with their source publications, and are excluded from this table. The uncertainties mentioned with the palaeo-water depth offsets are used as the IR uncertainty.

Table 1 lists standard values for water depth and associated uncertainty per peat type, as well as how this relates to ULD, SLIP or LLD classification. The latter classification is determined by evaluating peat bed thickness, sample position (Sect. 3.3.1), bed lithology, botanical composition (Table 1), and further considerations of the stratigraphic and geographic position of the sample (setting) – usually with some iterative cross-checks (Hijma and Cohen, 2019). First, all basal peat samples are regarded as groundwater index points (RWL = GWL). Second, the group is divided into a seaward and shallower/younger subset, and an inland and deeper/older subset, based on geographic location and age-depth information as a starting assumption, further improved by iterative comparison with surrounding age-depth data. For the seaward, younger/shallower subset, the GWL can be regarded as controlled by tidal waters and, therefore, relatable to sea-level (RWL = GWL = MHW; Van de Plassche, 1995a; Shennan et al., 2000). For the inland, deeper/older subset, GWL is considered otherwise controlled and to have been positioned well above contemporary MHW levels (Van Dijk et al., 1991; Van de Plassche, 1995a; Cohen, 2005; Vis et al., 2015; Hijma and Cohen, 2019).

Table 1 Indicative meanings for various peat and organic facies for ULD, SLIP and LLD entries (after Hijma and Cohen, 2019).

| Sedimentary indicator facies 'peat type' (Field 54) | # entries (N=670) | Palaeo-water-depth [m] | Sample indicative meaning (Field 56) | | | IR uncertainty (Field 58) |
| --- | --- | --- | --- | --- | --- | --- |
| | | | ULD vertical positioning | SLIP vertical positioning | LLD vertical positioning | Palaeo-water-depth uncertainty [m] |
| Bog peat | 5 | 0 | GWL | | | ± 0.1 |
| Fen-wood peat | 249 | 0 | GWL | MHW | | ± 0.1 |
| Fen peat on inland dune flanks | 56 | 0 | GWL-0.1 | MHW-0.1 | | ± 0.2 |
| Fen-bog peat | 2 | 0.3 | GWL-0.3 | MHW-0.3 | | ± 0.2 |
| Fen peat | 167 | 0.3 | GWL-0.3 | MHW-0.3 | | ± 0.2 |
| Undifferentiated peat types | 145 | 0.2 | GWL-0.2 | MHW-0.2 | | ± 0.3 |
| Gyttjaic organic beds | 24 | 0.75 | GWL-0.75 | MHW-0.75 | | ± 0.5 |
| Organic detritus/clay | 8 | 0.75 | GWL-0.75 | MHW-0.75 | MSL-0.75 | ± 0.5 |
| Other (includes: palaeosols, other drowned surfaces) | 14 | other | | | | other |

### 3.3.2 Base basal peat, Top basal peat

As introduced in Sect. 2, SLR during the Middle Holocene caused a concurrent rise of the coastal groundwater levels up to tens of kilometres landward (Jelgersma, 1961; Van de Plassche, 1982; Van Dijk et al., 1991; Cohen, 2005; Koster et al., 2017). This caused zonal paludification (i.e., extensive peat growth) of the Pleistocene subsurface underlying the eventual Dutch coastal plain. The so-called basal peats that formed this way are of variable botanical composition (Bennema, 1954; Van de Plassche, 1982; Cohen, 2005; Bos et al., 2012). The very base of the peat bed overlying the Pleistocene substrate (Figure 3b) is regarded to mark the beginning of peat formation: the organic facies reflect that year-round swampy to marshy conditions have established at that location, and radiocarbon dating of these facies reflects when this occurred. Together, the age-depth data thus pin a past GWL position that in river mouth and lagoon rim situations, in turn, provides an upper limit to the sea level position of that time (Berendsen et al., 2007; Hijma and Cohen, 2010, 2019; Van de Plassche et al., 2010; Koster et al., 2017; Meijles et al., 2018; Quik et al., 2022). This base-basal peat sample context and index-point use concept applies to 622 of the 640 basal peat data points (incl. 105 SLIPs, 337 tidal ULD), dated at the base or in the middle of the peat bed.

A variant is to date samples from the very top of a basal peat bed where it is overlain by tidal clays (Figure 3b), preferably in addition to dating the base of the peat bed. This then provides a second age-depth water level index point, more directly marking the marine inundation of a young peatland surface just above the older subsurface. This sampling context and index-point concept applies to ~20 of the 640 basal peat data points. Decompaction offsets and uncertainties (see Sect. 4) are larger for these top basal peat points, which propagates in the vertical accuracy of the water level index point (e.g. Makaske et al.,

2003; Berendsen et al., 2007; Hijma and Cohen, 2019). The indicative meaning based on peat composition (Table 1) is assigned indifferently to base, mid and top sampling.

### 3.3.3 Data points from peat beds on inland dune flanks

Section 2 introduced inland dunes as part of the Rhine-Meuse and Overijsselse Vecht palaeovalley substrate (Fig. 1a). These dunes formed in the Late-Glacial and at the start of the Holocene (15-10 ka cal. BP) and after an interlude marked by the formation of a palaeosol, were gradually covered owing to Holocene water level rise, peat growth and coastal-deltaic sedimentation. Along the swamp and fen-rimmed dune flanks, local peat formation often could keep up with the water level rise. This has created particularly favourable sites to collect age-depth data series that span several meters of elevation

difference at a single location, while still meeting the condition that the sampled peat bed is on the relatively compaction-free substrate (Figure 3b). For this reason, many inland dunes have been sampled to reconstruct past water levels (Jelgersma, 1961; Van de Plassche, 1982; Törnqvist et al., 1998; Berendsen et al., 2007; Van de Plassche et al., 2010). Especially the outer flanks of the highest dunes in inland dune complexes are suitable for sampling. In lagoon and lower deltaic settings, a subset of these dune flank samples can be upgraded from a GWL-related to MSL-related indicative meaning (Van de Plassche, 1982; Van de

Plassche et al., 2005, 2010; Hijma and Cohen, 2019). Sampling bases of basal peat from near the base of the dunes, where the flanks are less steep and base topography becomes uneven, results in age-depth data points less suitable for relating to MSL. Such locations provide ULDs that, with further analysis, often prove to be perched above contemporary basal peat GWL of the surrounding floodplain.

Besides dated peat samples from inland dune flanks, the HOSLEA-NL data set contains 22 non-basal peat [14]C-dated charcoal ULD entries that also come from dune flanks. They are from swamp rim archaeological beds at the location where these intersected elevation contours of inland dunes (Van der Woude, 1983; Verbruggen, 1992). The archaeological bed surface is overlain by peat, marking the paludification of the area. The charcoal dates from just underneath constrain the timing of this, thus providing a GWL age-depth point.

**3.3.4 Data points from peat beds on palaeovalley floodplain surfaces**

Section 2 introduced palaeovalleys as low elevated corridor areas with relative early and extensive basal peat growth and as gateways to further transgression (Figure 1a; Figure 3a). Their substrate consists of terraced fluvial sands, topped by a consolidated sandy clay floodplain unit that bears a developed palaeosol (Rhine-Meuse palaeovalley, widespread: Wijchen Member cf. Törnqvist et al., 1994; Autin, 2008; Overijsselse Vecht palaeovalley, more locally: Singraven Member). Basal

peat overlying the floodplain surface associates with a river-flooding hydrological regime, characterised by eutrophic conditions and abundant reed, wood and fen-woody peat types (Bos et al., 2012). The pre-consolidated state of the underlying floodplain deposits at the time the peat formed, as marked by the palaeosol features, makes it a relatively compaction-free substrate, giving it a similar vertical accuracy as for basal peat index points from inland dunes and coversand (Cohen, 2005;

Koster et al., 2017). Basal peat formed most extensively in relatively distal parts of the palaeovalley, i.e., in areas flooding

regularly but receiving relatively little flood sediment. Peat started forming when downstream sea-level rise initiated rising floodplain groundwater tables to just above the palaeosol surface, transforming floodplains into year-round moist flood basins, with the groundwater tables and their gradient coinciding with mean-annual river water levels (Van Dijk et al., 1991; Kiden et al., 2002; Cohen, 2005; Koster et al., 2017). Along the rims of the palaeovalleys, GWLs indicated by basal peats can be more perched due to local seepage hydrological conditions (Cohen, 2005; Bos et al., 2012). The river gradient and seepage overprints

make that those base basal peat data points cannot be related to MSL and provide for inland GWL age-depth points only, in some cases plotting meters above the contemporary sea level (Cohen, 2005; Koster et al., 2017; Hijma and Cohen, 2019). Data points evidently affected by these overprints are classified as secondary ULD types (ULD river gradient; Fig. 2), based on age-depth position comparisons with contemporary index points from downstream sites.

Top basal peat age-depth points from palaeovalley settings do not suffer from these river gradient effects and have stronger relevance for sea-level reconstruction. Where the peat is non-erosively overlain by fluvial-tidal (Rhine-Meuse palaeovalley) or fully tidal deposits (Overijsselse Vecht palaeovalley), the age of the top of the peat indicates the time of flooding by the sea and thus yields a SLIP (Hijma and Cohen, 2019). This is the case in the seaward parts of the palaeovalleys, where the river valley turned into an estuary. Here, transgression commenced relatively early and rapid and eventual inundation was many

meters deep under brackish tidal open water regimes (Figure 1; Figure 3a).

### 3.3.5 Data points from peat beds on coversand relief

Outside the palaeovalleys, basal peat is predominantly encountered overlying coversand relief (Figure 3c). This buried surface often contains a developed podzolic palaeosol in the aeolian sand, indicating regionally low groundwater table positions during

and after aeolian deposition (up to 12 ka cal. BP) and prior to basal peat formation (gradually after 9.2 ka cal. BP; Fig. 2b). Basal peat on coversand often features a transitional contact from palaeosol organics (peaty sand) to the actual peat bed, regarded to be a result of relatively gradual drowning (Van de Plassche, 1982; Cohen, 2005), similar to the signal in floodplains and along inland dune bases in the palaeovalleys. Base basal peat samples from coversand terrain contain ULD, especially when obtained from along flanks of subdued dune relief (Sect. 3.3.3). Again, top basal peat samples form potential SLIPs,

provided that the top is non-erosively overlain by tidal sediments and that the thickness of the peat layer is limited.

In contrast to the eutrophic swampy basal peat on floodplain deposits and inland dunes in palaeovalleys, where the vicinity of flooding rivers brings nutrients, the peat overlying coversand often has a mesotrophic composition (sedge peat types). This is indicative of seepage-dominated hydrological regimes and explains the relatively perched positions of coversand basal peat

data points, compared to nearby contemporary palaeovalley data sets (Van de Plassche, 1982; Cohen, 2005). The degree of seepage evidence strongly echoes local coversand topography (Bos et al., 2012). Base basal peat dates from relative lows in

the coversand terrain tend to return relatively old dates, in which case they are regarded as influenced by local hydrological conditions. In the database these samples are categorised accordingly as ULD local GWL. Mesotrophic peat over coversand also dominate the lagoonal and tidal marsh fringing peat land formed since 5.0 ka cal. BP (Fig. 2d) north of the Rhine-Meuse

delta in the east-central Netherlands along the edges of the ice-pushed ridges and the Drenthe plateau, both source areas for regional seepage.

Overall, basal peats formed under freshwater conditions. Towards the top of these peat beds, reed-dominated layers often show evidence of brackish storm surges, identifiable by clayey deposits rather than shifts to salt-tolerant vegetation. The mud layers

overlying the top of the basal peat usually indicate the change to permanent brackish conditions. Reed is a plant species tolerant to a broader range of conditions and may grow under slightly brackish and mud accumulating conditions (Bos et al., 2012). However, where reed formed basal peats in the transgressive setting of the Netherlands during the Middle Holocene, it is not considered a brackish peat. Bos et al. (2012) found some sites with marine shells in gyttja deposits in the westernmost part of their study area, indicating that some organic layers formed under brackish conditions. During the Middle and Late Holocene,

reed stands along river mouths expanding into lagoons may have accelerated the transition from brackish, shallow water to fully terrestrial conditions. This process explains succession patterns within intercalated peat layers, as showcased for the Old Rhine mouth by Pierik et al. (2023), using, amongst others, diatom analysis. Such settings, however, do not apply to the shorter-lived basal peats in our database, in which brackish peats are not distinguished.

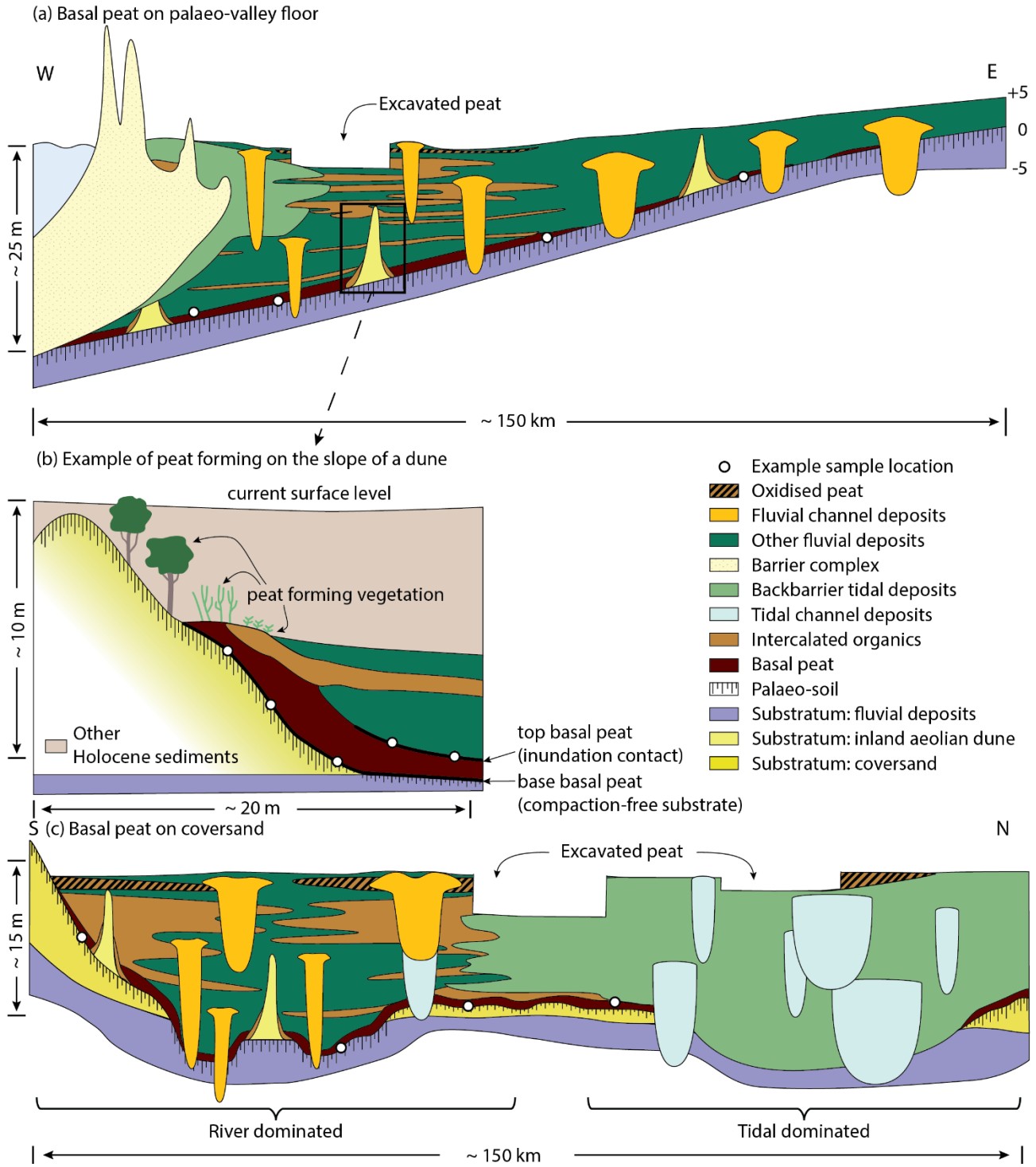

**Figure 3 Examples of different sedimentary locations of basal peat layers: (a) On the floor of a palaeovalley, (b) Along the flank of a river dune, (c) On top of coversand (adapted from Kiden et al. (2008) and Bos et al. (2012))**

### 3.3.6 Non-basal peat geological sea-level data

Some additional geological sea-level markers are reported in the HOLSEA-NL data set, including a set of brackish molluscs (Cerastoderma) obtained at great depth at Velsen (Van Straaten, 1954), and a set of index points obtained from [14]C dates of
in-viva shells (Scrobicularia plana; Cerastoderma edule) found in the top of tidal flat deposits underneath artificial dwelling mounds, or terps, in the northwest of the Netherlands (Vos and Nieuwhof, 2021). The older set from Velsen is used mainly as additional dating control to peat radiocarbon dates collected from the same excavation, which provide base basal peat and top basal peat data.

The younger set from the northwestern Netherlands is included as they supplement and cross-validate basal peat data in the younger period (3000-1800 years cal. BP) for which basal peat data is sparser (Sect. 2). The levels with the Scrobicularia plana shells are located just below the transition between tidal-flat and salt-marsh deposits. The vertical position of this transition is regarded to mark the past MHW, giving the shells an indicative meaning of MHW − 0.33 m (Scrobicularia plana) or MHW – 0.4 m (Cerastoderma) (Vos and Gerrets, 2005; Vis et al., 2015; Vos and Nieuwhof, 2021). Calibrated ages for these samples
are obtained using a marine calibration curve (Marine20; Heaton et al., 2020), and are corroborated by archeological dates from the overlying mounds. Because a regional offset for the reservoir age (delta R) is not established for the study area (W-NW NL), Marine20 was used without an additional delta R, as in Vos & Nieuwhof (2021). We note that this applies to the 12 Late Holocene shell dates regarded as SLIPs from the Wadden Sea (Vos & Nieuwhof, 2021), and to the six LLD shell dates and one tidal ULD from a North Sea incursion into the Overijsselse Vecht palaeovalley, away from direct exchange with Rhine
waters (Middle Holocene, Van Straaten (1954)). Including other shell dates from incursions into the Rhine-Meuse palaeovalley was not attempted because of unresolved delta R issues for the Rhine-Meuse mouth.

### 4. Age-depth positions and evaluation

Where Sect. 2 and 3 are restricted to data intake, setting diversity (local and regional) and overall classification goals (ULD vs SLIP), Sect. 4 describes further processing of the data in the HOLSEA-NL workbook. This comprises a series of adjustments
regarding the vertical position of an indicator, starting from the originally reported sample elevation and indicative meaning, incorporating further compiled information. This step essentially reproduces earlier similar applications on subsets of the dataset (Kiden et al., 2002; Berendsen et al., 2007; Hijma and Cohen, 2010, 2019; Van de Plassche et al., 2010; Hijma et al., 2015), more uniform and over the entire HOLSEA-NL data set. HOLSEA-standard fields were used to transparently register this and expanded where needed – also to allow omitting or substituting specific corrections in future usage. Addressing the
vertical corrections component-by-component serves this purpose. In many cases, the applied adjustments reproduce those from previous studies within a few cm, the difference owing to unification. This extends to acceptance/rejection and ULD/SLIP-usage decisions per data point, based on the age-depth data, indicative meanings and reliability judgements (e.g. Van de Plassche, 1982, fig. 4).

Importantly, the HOLSEA-NL data set also includes all adjustment specifications for originally demoted/rejected samples so that each decision can be re-evaluated per sample. Additional annotations on this are in note fields. This extension of the HOLSEA workbook improves the usability of the workbook and data set and is of particular relevance when evaluating the type of indicator a sample represents. For example, the RSL value of SLIP and ULD samples include the vertical adjustment for tidal amplitude to plot palaeo-MSL, and the palaeo-MHW is provided for all samples that are graded to the palaeo-MHW.

Otherwise, only the standard palaeo-GWL is documented. Documenting the palaeo-GWL enables the use of the dataset for relative water-level rise reconstructions throughout the Holocene coastal plain and its subsectors, and for spatial-temporal analysis thereof (e.g., sea-level history, subsidence regime, accommodation attribution, coastal prism architecture).

## 4.1 Vertical corrections

Vertical positions are specified and accounted for through (i) original sample depth + uncertainty, (ii) offset calculations to get

from sample depth to palaeo-water level depth, typically expressing GWL (Sect. 3), followed by (iii) further offset calculations to correct for compaction (upgrading depths), include palaeo-tides (upgrading to MSL expression) and corrections for two types of subsidence (upgrading depths) and (iv) error propagations associated to all of this. Figure 4 below shows the full set of components considered, and the order in which correcting vertical offsets are applied and evaluated. For all data points, several concurrent palaeo-water-level elevations can be calculated, as shown in Figure 4 and Figure 5. The systematic

processing chain facilitates iterative evaluation and switching between ULD and SLIP indicator type status (former GWL/MHW resp. MSL expression).

Decompaction (a vertical offset + uncertainty) is a correction that is always considered. Palaeo-tidal considerations are relevant to all data points that are classified as SLIPs and ULD, and the user may either use the provided values (expansion of standard

HOLSEA protocol; Hijma and Cohen, 2010, 2019) or fall back to using the modern tidal range in the region. The correction for background basin subsidence we regard optional to include. It is typically required (e.g. Kiden et al., 2002; Vink et al., 2007) if a comparison with GIA models is sought, though it is typically skipped (Hijma and Cohen, 2010, 2019; e.g. Van de Plassche et al., 2010) in traditional relative sea-level curve construction plotting. The correction for so-called deep anthropogenic subsidence applies to specific subareas in the NE Netherlands that have been subject to 20-21$^{st}$ century gas

extraction and can be regarded as a sample depth correction (up to 0.4m) to apply in most use cases. Further details on these critical corrections are given in Sect. 4.1.1 to 4.1.4.

The application of the vertical corrections generally results in an upward shift of the final GWL or final RSL (pRSL in Figure 5) elevation with respect to the original sample elevation (Figure 5). On average, the final RSL elevation of basal SLIPs (n =

117) is 0.4 m higher than the original sample elevation. For intercalated SLIPs (n = 4), the difference is much higher (~1.5 m) due to the larger upward correction. The tidal correction strongly influences the elevation of the sample. Figure 5 shows that

assuming modern tides instead of using a palaeotidal model results in much lower RSL elevations, ~0.2 m below the original sample elevation. For Middle Holocene SLIPs, using a palaeotidal model causes an average upward shift of ~0.7 m because of the gradually increasing tidal range and tidal dampening. This shift is also present in Late Holocene SLIPs from areas with presumed strong tidal damping (Flevoland region), whilst in areas without tidal damping, the Late Holocene MHW is equivalent to modern MHW (Waddenzee regions).

### 4.1.1 Decompaction

Decompaction corrections can refer to the correction of two processes: (i) post-depositional compaction of the beds directly underlying the sample, which is mostly relevant for top-of-basal peat samples and can be up to 5 m and (ii) the self-compaction of the sampled material. This is a smaller component, but it applies to all peat samples, with the correction averaging 0.06 m. The post-depositional compaction is corrected for using the depth to the consolidated substrate (m) (Field 18 in supplementary HOLSEA workbook) and a decompaction factor. The decompaction factor depends on the overburden thickness (Field 17), with peat beds deeper below the surface experiencing more compaction than the shallower peat beds. For samples taken deeper than 20 m below the surface (n = 21), a decompaction factor of 3 is used. A decompaction factor of 2 is used for samples shallower than 2 m below the surface (n = 119). For all other samples between 2 and 20 m, a decompaction factor of 2.5 m is used. This is documented as the compaction correction (m) (Field 64) in the HOLSEA workbook. We present an example from the first category to demonstrate how the decompaction factor affects the compaction correction, expressed as an offset to sample depth. A decompaction factor of 3 implies that, at the time of inundation, the peaty layer between the sample and the Pleistocene substrate was three times thicker than its current thickness (T), recorded as "Depth-to-consolidated-surface" in Field 18. Therefore, the upward offset stored in Field 64 should be 3 times the current elevation above the consolidated surface, which equals 2 times the thickness from Field 18 (T + 2T = 3T; thus, 3T – T = 2T). For decompaction factors of 2 and 2.5, the multiplier used in Field 64 is 1 and 1.5, respectively.

In specific cases, when land was reclaimed and the current land surface is below modern MSL, e.g. 'deep polders', the decompaction could be underestimated since the current overburden thickness is lower than before reclamation, which is currently not accounted for. The associated uncertainty is compaction correction dependent (Field 65). The compaction correction uncertainty considers an error margin of 0.02 m for the depth to the consolidated surface and assumes an uncertainty of 0.5 in the decompaction factor ($\sqrt{(\frac{0.02}{Field\ 18})^2 + \frac{0.5}{decompaction\ factor})^2} * compaction\ correction\ (Field\ 64)$). Thus, for a sample with a midpoint 0.10 m above the consolidated substrate and taken 8 m below the surface, the decompacted midpoint is 0.25±0.07 m above this substrate (with 0.25 = 0.10 + 0.15 = Field 18 + Field 64).

For the sample thickness (Field 21), a decompaction factor of 2.5 is applied around the sample midpoint, consistent with investigations in Van Asselen (2011) and past usage in Hijma and Cohen (2019). The sample decompaction uncertainty (Field 24; 'Sample thickness uncertainty (m)') is set to half the decompacted sample thickness. This is a modification of earlier decompaction approaches by Berendsen et al. (2007) and Van de Plassche et al. (2010) who used a decompaction factor of 2.5 for the bases of their sampled beds and 3.5 for their tops, accounting for both the compaction of the underlying unconsolidated sediment as well as the compaction of the sample itself. In the HOLSEA format, these two components are split into separate steps, which is why a single decompaction factor was used to correct the sample thickness relative to the mid-point and a separate decompaction factor for decompacting the underlying unconsolidated sediments. This also allows us to account for the larger variations in sample depths encountered in the HOLSEA-NL data, compared to the smaller variations in depths in the studies by Berendsen et al. (2007) and Van de Plassche et al. (2010), which focused on specific sites.

### 4.1.2 Past Tidal Amplitude and Flood-basin Effect

A tidal amplitude is required to calculate the vertical positions of SLIPs that express MSL. This counts for data points that have a RWL = GWL = MHW assigned indicative meaning (Sect. 3.3.2). The tidal amplitude (half the tidal range) is the offset between MHW and MSL and unlike all other vertical corrections considers a downward vertical correction (Fig. 4 and 5). Modern tidal ranges along the Dutch coastline are controlled by the North Sea bathymetry and its connection to the ocean's global tides. Recent variability of these tidal ranges is known from centuries of observations and the ranges are regarded to be relatively stable. The modern tidal circulation is considered to have established when global sea levels reached their high stand and the North Sea approached its modern depths, from around 6800 years ago (Van der Molen and De Swart, 2001). Over the entire Holocene, however, the tidal ranges are considered time-variant, especially for the period from 9000 to 6800 years ago when the southern North Sea was inundating, shallow bathymetries deepened, and coastline positions shifted. To allow for this in the HOLSEA-NL set, rather than using modern tidal amplitudes (HOLSEA standard), we implemented a lookup scheme for palaeo-tidal amplitudes (available from earlier model-reconstruction work; Van der Molen and De Swart, 2001; Uehara et al., 2006; Ward et al., 2016), and used those for specifying the tidal corrections per sample. Herein, we geographically expanded an earlier palaeotidal-correction application by Hijma & Cohen (2010, 2019). As in Hijma & Cohen (2019), we specified the palaeo-tidal amplitudes (pMHW) as meta-information in note fields in the HOLSEA workbook. This considers MHW (pMHW) in the near shore, just seaward of the modern coastline.

Nearshore tidal amplitudes propagate inland and are deformed into estuarine, fluvial-tidal, and lagoonal waters inshore, to which the peat-forming hydrological systems originally graded. In the wide, underfilled, estuarine back-barrier lagoonal-deltaic and lagoonal situations of 9000-5000 years ago (Fig. 1), inland tidal dampening is regarded to have occurred in SLIP producing areas (Hijma & Cohen, 2019). This inland lowering of water tables to a level in between seaward MHW and MSL is called the flood-basin effect (FBE) and occurs especially where the semi-diurnal tidal wave travels through a narrow bottleneck in and out of broader flood basins inland (Van Veen, 1950; Zonneveld, 1960; Van Veen et al., 2005). To implement

this additional tidal correction, used as a modifier of the values provided by the pMHW geographic lookup scheme, the dataset specifies an FBE-factor per sample.

Various studies have assessed past FBE in RSL-reconstructions, in particular in the lower Rhine-Meuse area and the Flevo-lagoon region (Van de Plassche, 1982, 1995a; Berendsen et al., 2007; Van de Plassche et al., 2010; Vis et al., 2015; Hijma and Cohen, 2019), by cross-comparison of age-depth plots for relative seaward, central and inland subsites. This way, the sites where GWL age-depth data plots the youngest-deepest can be identified and from this, it is inferred that the flood-basin effect must have been largest at these sites. These youngest-deepest points are promoted to SLIP status (more directly related to past MSL positions) while leaving surrounding contemporary sites as ULD. Assessment whether flood-basin effect was intermediate (some dampening) or maximally developed (full dampening) requires cross-checks with palaeogeographical reconstructions. Hijma and Cohen (2019) did the latter for the lower Rhine-Meuse delta, leading to a prescription of past FBE as intermediate (50% dampened, FBE = 0.5) in the relative open situation before 7.5 ka, FBE = 0.75 between 7.5 and 6.5 ka, and full dampening FBE = 1 established 6.5-3.0 ka. For the other regions, FBE corrections for data points classified as SLIP were newly assessed, based on the HOLSEA-NL data coverage, with cross-checks against available palaeogeographical reconstructions (Vos et al., 2018; panels in Fig. 2 and additional), especially in the Flevo lagoon and northern regions. For the western coastline, North Holland to Zeeland, we prescribe the same tidal dampening regime as used by Hijma and Cohen (2019) for the lower Rhine-Meuse delta. For the Waddenzee region, tidal dampening is assumed to have been less prominent. The indicators from the western Waddenzee are assigned no flood-basin effect since no large flood basins formed around the SLIP producing samples. The eastern Waddenzee region is expected to have had some tidal dampening during the Middle Holocene, but this effect decreased towards the Late Holocene. Therefore, before 7 ka the tide is 50% dampened (FBE = 0.5), between 7 and 5 ka this decreases to 25% dampening (FBE = 0.25) and after 5 ka we assume no dampening.

We note that the prescription of FBE applies to the SLIP-producing locations and primary ULD. . The GWL at ULD locations surrounding SLIP locations is to be regarded as more strongly pMHW-controlled towards the sea and more strongly river-gradient or otherwise terrestrial hydrologically controlled inland, but the database does not quantify how strongly. The FBE correction factor is registered together with the pMHW value in the note field.

### 4.1.3 Long-term background basin subsidence

Because the coastal plain of the Netherlands overlies the long-term sinking North Sea sedimentary basin, it should be considered that part of the relative water level rise documented by the dataset is due to tectonic and sedimentary loading subsidence. Therefore, optionally, depending on application (see above), the effect of this background term is removed by applying a vertical correction. As input to this correction, a map product specifying a background rate was used, following the approach of Cohen et al. (2022: their Sect. 3.3 Vertical Land Motion) in their Last Interglacial sea-level database. This map considers estimates of long-term mean subsidence rates calculated over the 1.8 Myr, derived from onshore and offshore

Quaternary basin fill mapping, along with an associated uncertainty specification. For more details see Cohen et al. (2022)
including their Sect. 6.6 on preferring 1.8-Myr averaged rates, which are 80-70% that of 2.6-Myr rates. The spatial patterns and values are consistent with earlier tectono-sedimentary back-stripping analyses for this region (Kooi et al., 1998; producing rates calculated over the last 2.6-Myr) and applications thereof in relative sea-level data analysis in Kiden et al. (2002), Vink et al. (2007) and Simon and Riva (2020).

For each sample location, values were read from the vertical land motion map product and multiplied with the sample's age, with the uncertainty on age plus rate propagated accordingly. The HOLSEA data workbook stores this under "Tectonic correction (m)" (Field 66) and "Tectonic correction uncertainty (m)" (Field 67). The upward corrections range from 0 up to 1 m in the study area, with uncertainties between 0.01 and 0.18 m. Within the study area, rates are highest in the northwest. Averaged values for offsets considered for SLIPs plotted in Fig. 2b (9.2-6.6 ka cal. BP) serve as an example: Rhine-Meuse
palaeovalley 0.5 m; Vecht palaeovalley 0.8 m; inland Flevo-lagoon 0.6 m; NE Wadden Sea 0.3 m.

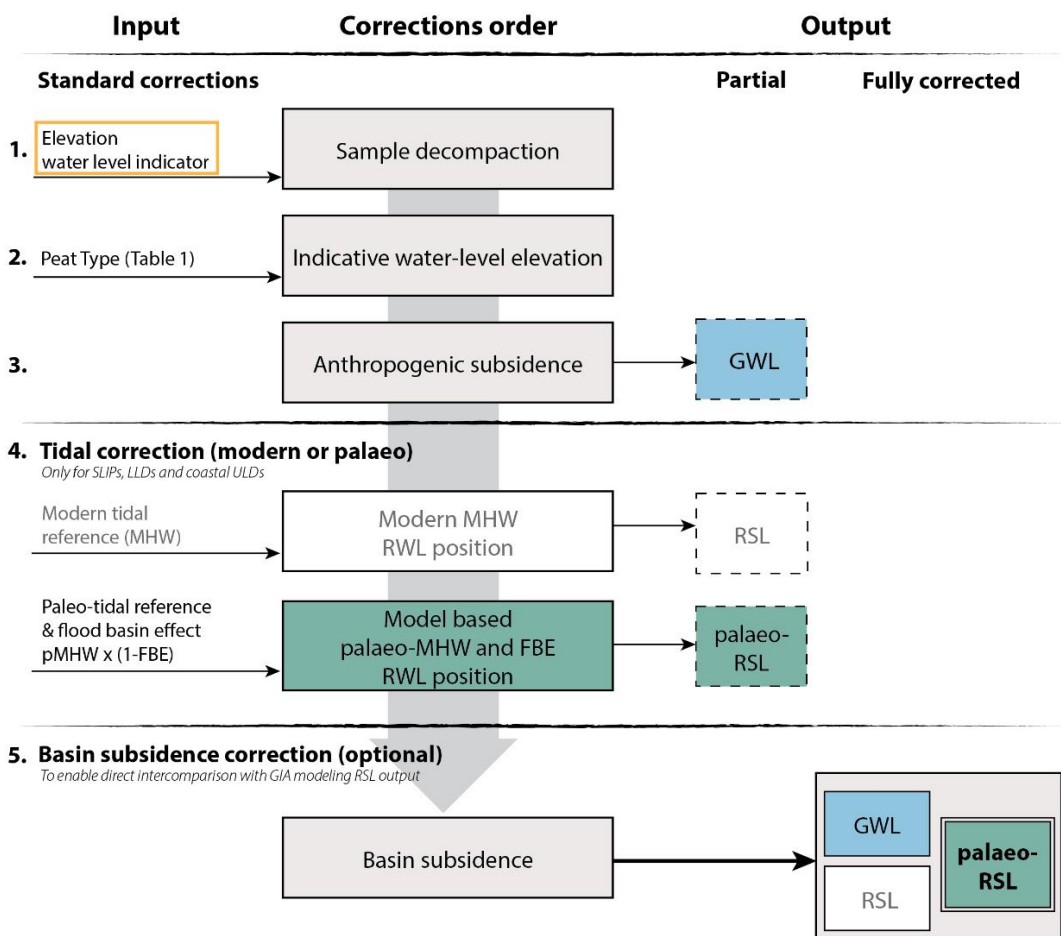

**Figure 4 Flowchart of water level indicator sample processing chain. The final fully corrected output is a basin subsidence-corrected palaeo-RSL value.**

### 4.1.4 Anthropogenic deep subsidence

The extraction of resources such as gas, water and salt from a range of depths well below the basal peats has caused significant recent subsidence in specific areas of the Netherlands (e.g. NAM, 2017, 2020). This subsidence has influenced the depth of Holocene water level indicators and an upward vertical correction is required to remove unwanted lowering of palaeo-water level indicators and correct joint plotting of samples collected 'in the 1950-60ies', 'in the 1990s' and 'in the 2010s' from these human-induced subsidence affected areas. This upward correction is dependent on the year of coring: larger for more recent years, and specified in the subsidence history maps by NAM (2017, 2020). In the strongest affected areas, this correction is up to $0.36 \pm 0.10$ m. The HOLSEA workbook records this separately as "Human-induced subsidence (m)" (Field 68) and "Human-induced subsidence uncertainty (m)" (Field 69).

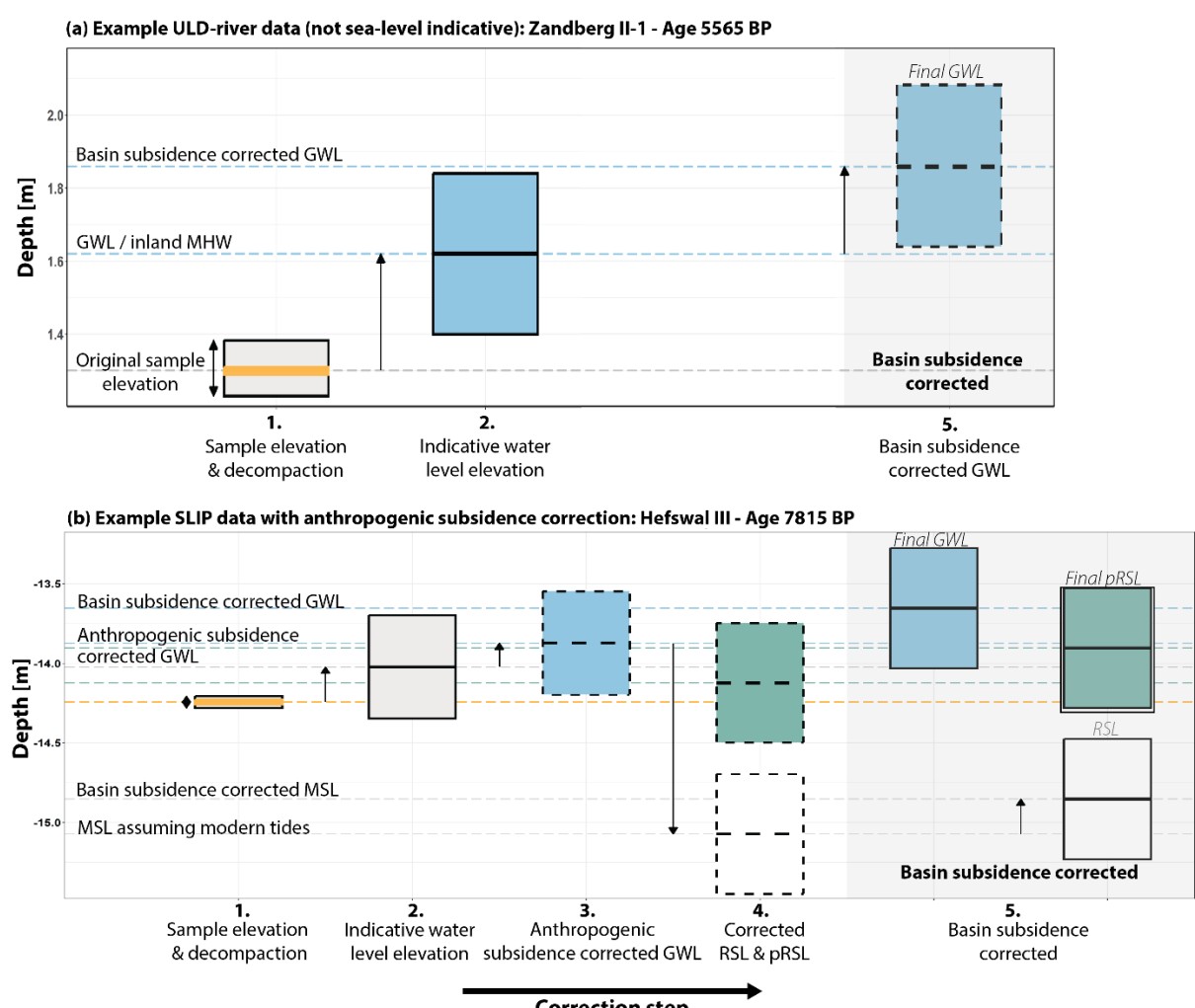

**Figure 5 Dashboard of sample elevations of two example samples for visual inspection of vertical positioning owing to vertical correction steps. (a) Inland GWL / Upper limiting sample without anthropogenic subsidence, (b) Fen peat SLIP sample from the northeast of the Netherlands, affected by deep anthropogenic subsidence**

## 4.2 Dating information, including calibration

Radiocarbon dating has been used to determine the age of all the samples. Because many data points were previously published and assigned various "unique sample IDs", we decided to use the lab code provided for dating as Unique sample ID (Field 1) to avoid confusion over the numbering system. Conform the HOLSEA workbook, sample ID (Lab-number), dating result (age + uncertainty, in $^{14}$C BP) are the primary dating information. Where available, $\delta^{13}$C (in ‰) and source reference of the date are provided as meta-information (see also Hijma et al., 2015; Hijma and Cohen, 2019). For very early radiocarbon dates measured before 1962 (original Gro-numbers from the Groningen lab), a later published correction for the Suess effect has been applied, following Vogel and Waterbolk (1963). These samples are documented in the HOLSEA-NL workbook using their converted

GrN-number, with the original $^{14}$C-age document in the notes. Additionally, a bulk error is provided for samples dated using conventional dating (e.g., GrN-numbers), except when explicitly stated that it was not a bulk sample (e.g., piece of wood). The bulk error is provided in the radiocarbon tab of the workbook.

The $^{14}$C-dating results from terrestrial material, such as basal peat macrofossils and charcoal, have been calibrated using the atmospheric calibration curve IntCal20 (Reimer et al. 2020). We recalibrated dates from older studies. Note that dating results earlier calibrated with IntCal04 and IntCal13 (in source literature) hardly differ from the IntCal20 recalibration (in HOLSEA-NL), as all our data is from within the Holocene part of the calibration curves.

### 4.2.1 Bayesian calibration

Besides individual date calibrations ('unmodelled' calibrated ages), the workbook provides a second set of fields to allow for storing Bayesian calibration results for a vertical series of samples from the same site, as advocated by Cohen (2005) and Hijma & Cohen (2019). The 'modelled' calibrated ages were generated running CQL scripts in the OxCal 4.4 software (Bronk Ramsey, 2008, 2009) partly reused from Cohen (2003) and Hijma & Cohen (2019). The sequential calibration model further narrows down the calibrated age of some samples, decreasing the age range by 10 up to 500 years. The decrease in age range

is generally larger for older samples that used conventional dating (mostly dated before 2000) compared to those dated using AMS-dating.

### 5. Processed data overview

This section summarises the dataset contents in its full processed form, highlighting the newly achieved uniform coverage. The focus is on describing systematic spatial differences and showing the quantitative effects of including regarded optional

vertical corrections. To do so, the data is grouped into seven regions (Figure 6), that are also Holocene-geologically different. The boundaries between coastal subregions follow Pleistocene drainage divides (Vos et al., 2011, 2018; Cohen et al., 2017). The Rhine-Meuse and Vecht transgressed palaeovalleys are each west-east subdivided based on dominance of coastal-and/or-tidal (seaward) vs. fluvial-and/or-peaty Holocene depositional circumstances (landward) (Figure 2).

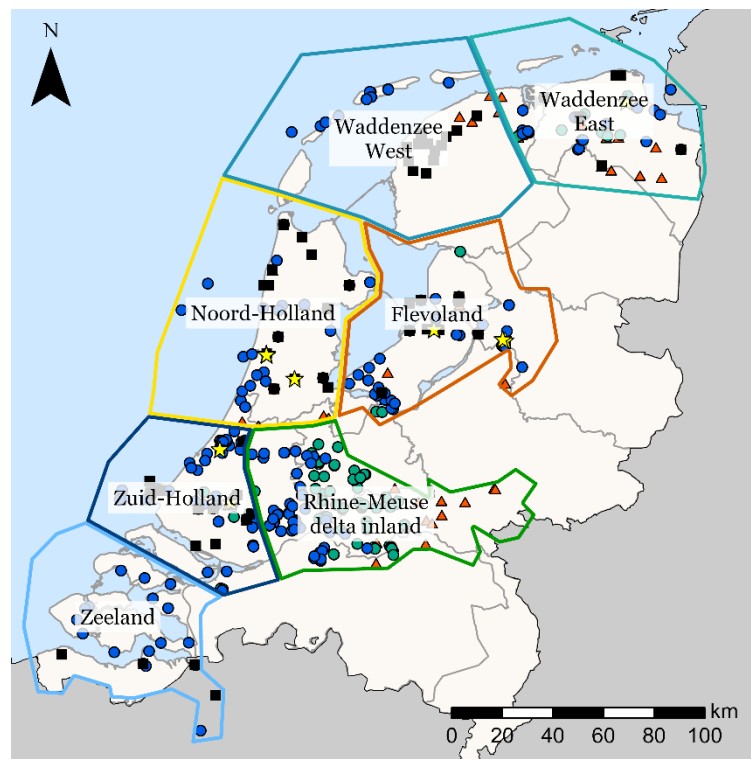

**Figure 6 Overview map of water-level indicators and region divisions. Multiple indicator points from the same location are plotted on top of each other.**

## 5.1 Spatial distribution

The HOLSEA-NL data set contains 682 water-level indicators, of which 121 SLIPs, 14 LLD points, 368 primary ULD points, 129 river-gradient affected ULD and 50 local-GWL secondary inland ULD data points. Most of the collected data is from the Middle Holocene (between 8.2 and 4.2 ka BP; Figure 7a) because of three main reasons. First, this period corresponds with the inundation of the Pleistocene surface in large parts of the Netherlands during the Middle Holocene, due to which large-scale peat growth was possible throughout the coastal areas of the Netherlands (Vos, 2015b). In this period, most of the basal peat layers were formed (Figure 2b, c). It can be seen in Figure 7a that the ULD local GWL indicators are the relatively oldest. In Early Holocene times, long before eventual coastal transgression, peat formed only locally because the sea level was not high enough to reach above the Pleistocene surface (majority of data older than 9.2 ka). Older basal peats related to RSLR are found offshore, mostly outside the study area, where the Pleistocene topography had a lower elevation, resulting in an earlier inundation. Conversely, the many younger basal peat samples (9.2 to 3.0 ka data) are found further inland or on top of Pleistocene covers sand and inland dunes. Second, erosion has caused the basal peat to disappear in areas, both along the coast and inland, resulting in the disappearance of many younger peat layers, causing the drop in general numbers after 5 ka. Third, in addition to natural erosion, human activity has resulted in the removal of large amounts of peat from the subsurface (Vos, 2015b). Large-scale peat excavations have occurred since the Middle Ages when peat was mined extensively (Pierik et al.,

2017). Additionally, embankments of rivers and artificial drainage led to the oxidation of peaty top soils. The result is that large parts of the shallower peat layers in the Netherlands have disappeared, including some of the younger and shallower basal peat layers. Therefore, water level reconstructions of the Late Holocene require additional methods and indicator types other than basal-peat samples.

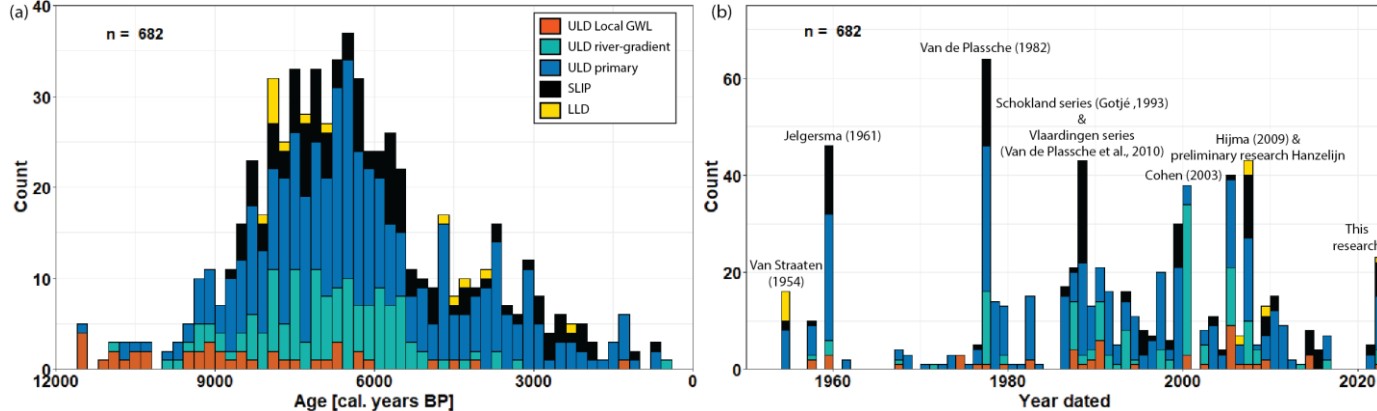

**Figure 7 (a) Distribution of the number of water level indicators by age from 12 to 0 ky cal. BP; (b) Distribution of water level indicators sorted by the year the samples were dated. Some large sampling campaigns are indicated. Rejected data is omitted.**

Figure 7b shows when the indicators were dated and serves as an illustration of the diverse, stepped research history behind the database contents. Many of the samples were collected and dated in large campaigns, some of which are noted in Figure 7b. From the '80s onwards, smaller sets (single sites) were collected, and the nature of the producing studies diversified. Since 2000, routine sampling as part of archaeological prospection attached to infrastructural projects (e.g., Hanzelijn railroad through Flevoland) has become an important supplier. The 2022/23 spike includes newly obtained dates from the North Holland-Flevoland fringe region collected as part of the present research while compiling the database (highlighted in Sect. 5.2; HOLSEA-NL primary reference: "this paper"; De Wit and Cohen, (2024)). An overview of all primary source references is given in Table 2.

When dividing the data over the regions, some distinction in year of dating is visible between the different regions. This variation shows the shifts in focus for water- and sea-level research in the Netherlands. For example, in Noord-Holland, most samples were dated already between 1955 and 1960 by Van Straaten (1954) and Jelgersma (1961), with very limited additional samples. We tried to fill this gap with new submissions in 2022-2023. On the other hand, from Flevoland most of the samples are collected from 1989 onwards, starting with the samples collected by Roeleveld and Gotjé (1993) (published in Gotjé (1993)). Figure 8 reveals the spatially uneven distribution of water level indicator collection, with the great majority collected in the central part of the Netherlands (Zuid-Holland, Flevoland and Rhine-Meuse delta inland). This is again strongly linked to the research focus of past studies. For the northern areas, the number of indicators is gradually increasing over the past two decades, showing an increasing interest in GWL and RSL reconstruction in this area (Meijles et al., 2018; Quik et al., 2021; Makaske and Maas, 2023). Especially for Zeeland in the southwest, the collection of new data is very limited.

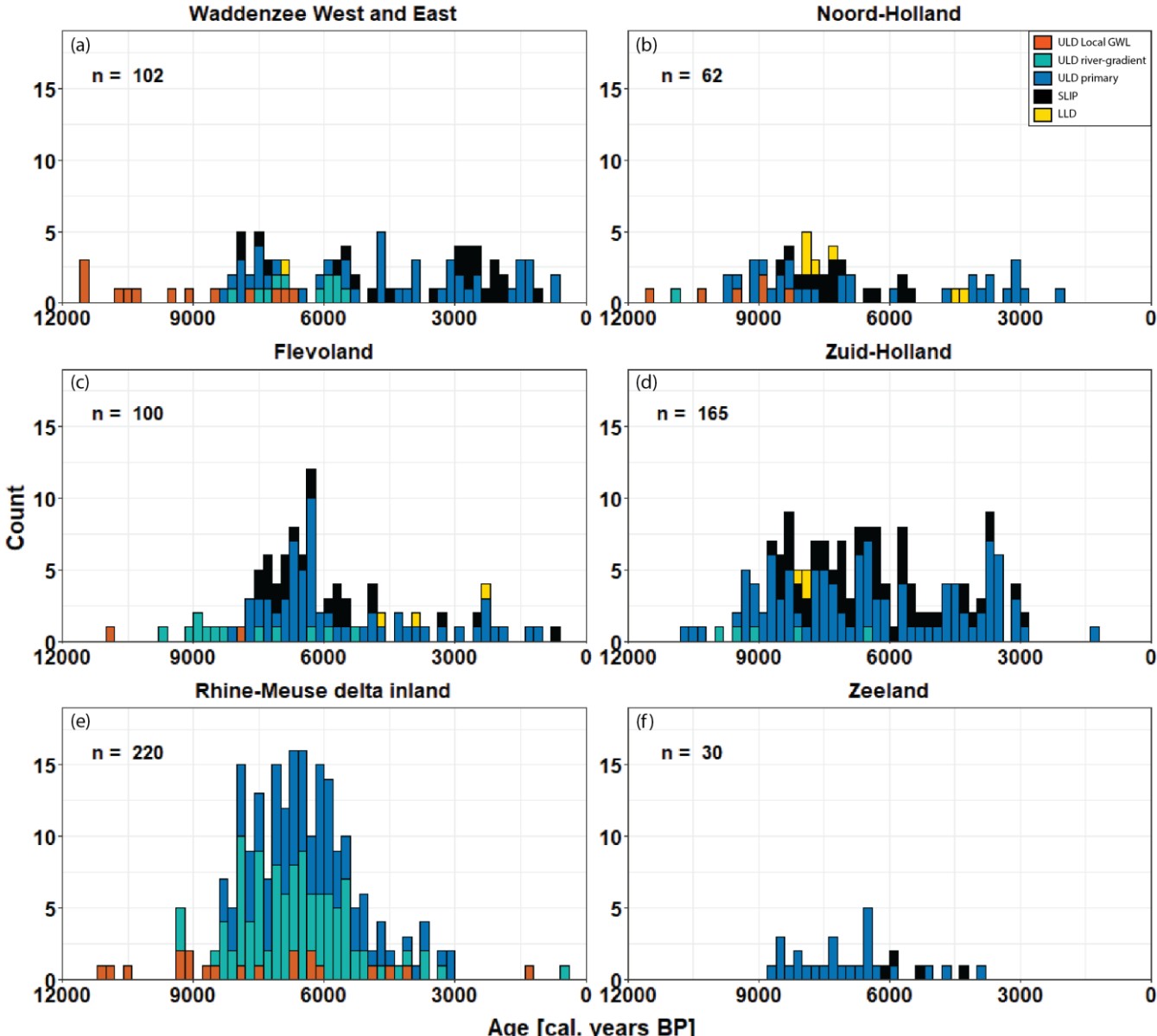

**Figure 8 Distributions of water level indicators sorted per subregion by age from 12 to 0 ky cal. BP. Rejected data is omitted.**

## 5.2 Age-depth plots

Separating the age-depth plots of the corrected water-level data (GWL, without palaeo-tidal correction) per region shows the spatial variability in the data (Figure 9). A consistent water-level rise trend is visible in the denser sampled regions Zuid-Holland, Rhine-Meuse delta inland, Flevoland and Waddenzee. In contrast, the plots from Noord-Holland and Zeeland show a patchier pattern because of the lower number of indicators in these regions. Regarding temporal coverage: the coastal regions

Zuid-Holland, Noord-Holland and the Waddenzee host relatively older ULD-tidal and SLIP data, starting ca. 9-8.5 ka cal.

years BP, while in the more inland regions, Flevoland and Rhine-Meuse delta, this commences ca. 8-7.5 cal. years BP. This reproduces and confirms earlier investigations of the transgression rate (e.g. Vos, 2015b; Koster et al., 2017).

The setup of the HOLSEA-NL workbook allows for several variants of age-depth plotting, useful at different stages of evaluation and iterative classification (SLIP, ULD etc.) and for different derived data set use (see also Sect. 4.1). The data set can be used to make customised age-depth plots using for instance original depth, inferred GWL position, further inferred MSL positions, or further background-subsidence corrected MSL positions (Figure 4 and Figure 5). Figure 10a-c provides three variants of indicator depth plotting against age that can be zoomed into to evaluate clusters of apparent outlying data for possible under- or overcorrection. To prevent misinterpretation of the different age-depth data, we recommend using explicit labelling and clear caption details.

Overall, SLIPs, ULD and LLD subsets each show relatively rapid water-level rise during the Middle Holocene (from ca. 9 ka onwards), which slows down towards the Late Holocene (semi-linear after 4 ka). The river gradient ULD echo this trend, but in higher positions, lifted up by the river gradient. It is also clear that the local GWL ULD does not follow the general curve but shows a more diverse pattern (Figure 10a). Figure 10a shows the elevation of all the water-level data through time with corrections for compaction, background basin subsidence and anthropogenic deep subsidence (corrected GWL), but without tidal corrections. Figure 10b shows the corrected water-level data (in grey) and the fully tidal corrected RSL data. The corrected water-level data shows a slight upward shift compared to the uncorrected data. Also, the propagated uncertainty to the depth positions increases, especially for the older data points.

To illustrate what the databasing activity has added to the inventory – and how zoomed-in age-depth plot evaluations worked out, we highlight the data newly added and assembled for North Holland and Flevoland. North Holland has been a region where basal peat data was relatively scarce (e.g. Van de Plassche, 1982; Koster et al., 2017), and Flevoland a region where basal peat data is of diverse origin (Schokland research efforts: Roeleveld and Gotjé, 1993; Van de Plassche, 1995a; Almere research efforts: Makaske et al., 2003; various archaeological investigations 2000-2020s). To add to Middle Holocene data coverage, a cluster of sites was dated from SE North Holland (Durgerdam, Slotermeer, Diemen: this research, N=12) and from within IJsselmeer (Van den Brenk et al., 2023; this research, N=5). To add to Late Holocene data coverage, dates from central Flevoland (Hanzelijn: Hamburg and Knippenberg, 2006; De Moor et al., 2009; N=16; Kampen-Cellemuiden, this research; N=5) were added to the existing data. To highlight some findings and actions:

-   Inspection of the Flevoland data after initial entry revealed outlier clusters 5000 to 4000 years old plotting 'too young, too deep', that in databases were registered as 'basal peat' dates but in detailed reports (Van Lil, 2008; De Moor et al., 2009) had actually been identified as peat detritus: sediment from peat-lake bottoms, calved from fringing peat bodies, redeposited at lower elevations. These data points were demoted to LLD data points, and they are partly rejected as sea-level indicators (see Figure 9c).

- Inspection of the remaining (non-lake bottom) Hanzelijn and the Kampen-Cellemuiden data allowed the identification of the lowest-youngest ULD data points in the series and, in three cases, upgrade these to SLIP status for c. 3000, c. 2500 and c 900 cal. yr. BP (Figure 9c), which supplement and confirm the non-basal peat terps SLIP set (see Sect. 3.3.5). The eastern fringe of the Flevoland coastal plain is identified as a region holding MSL-reconstruction relevant peat resources at a relatively shallow position (within 1 meter below OD), more strongly and explicitly than in earlier national compilations.

- Inspection of data from SE North Holland for location Slotermeer at 7500 cal. yr. BP, identified it as a 'too young, too deep' outlier cluster: presumably a Middle Holocene lake situation, similar to the Late Holocene 'Hanzelijn' example mentioned above. These data points were demoted to LLD data points (see Figure 9b).

- Inspection of potential onset Late Holocene SLIPs from locality Schoorl in NW North Holland (4500-3500 cal yr BP, (Ente et al., 1975)) suggests these to plot relatively low (Figure 9b), for which there are competing explanations: tectonic correction (~0.45 m over 4000 yr) may be locally underestimated, tidal correction (0.8 m) overdone, or compaction correction (0.15 m) underestimated. Because of the uncertainty in the vertical corrections and the low elevation of the samples, these potential SLIPs were rejected. Additional sampling in this area and more detailed research on the vertical corrections at this site will improve the age-depth reconstruction and help identify why these Schoorl points are plotting lower than surrounding points.

A selection of the SLIPs per region from the parent database can be used to fit relative sea-level curves per region. Such curves will deviate back in time because of differential subsidence – N Netherlands curves being positioned below the SW Netherlands curve for the period between 8.5 to 5 ka cal. BP. For sea-level markers of ca. 8000 years ago, data (with tectonic-correction applied) and tentative curves in Figure 10c suggest 2 to 4 m more subsidence to have occurred in the N Netherlands than in the SW. This is a similar finding as earlier communicated – partly corroborating, partly reproducing, partly detailing it – in work by Kiden et al. (2002) and Vink et al. (2007), who attribute the regional differences in the Netherlands to GIA related differential subsidence.

A more extensive comparison of the trends from different regions is recommended, for example, using Bayesian modelling for the SLIP data, like in Cahill et al. (2015). Narrower quantification of the GIA contribution and its decay from the Middle Holocene (8000-4000 cal. yr. BP) to Late Holocene (last 4000 years) using the HOLSEA-NL database is part of ongoing research. This type of research extends from geological data to incorporating modern tide gauge, GNSS and satellite sea-level monitoring era data (e.g. Vermeersen et al., 2018; Steffelbauer et al., 2022) – beyond the scope of this dataset and paper that is restricted to geological water level data. Nevertheless, evaluating the sea-level rise and subsidence rates from both realms, geological and modern, and integrated usage of the two resources is relevant when reconstructing the Late Holocene and recent RSLR (Simon and Riva, 2020), and the HOLSEA-NL database contributes to iterating that.

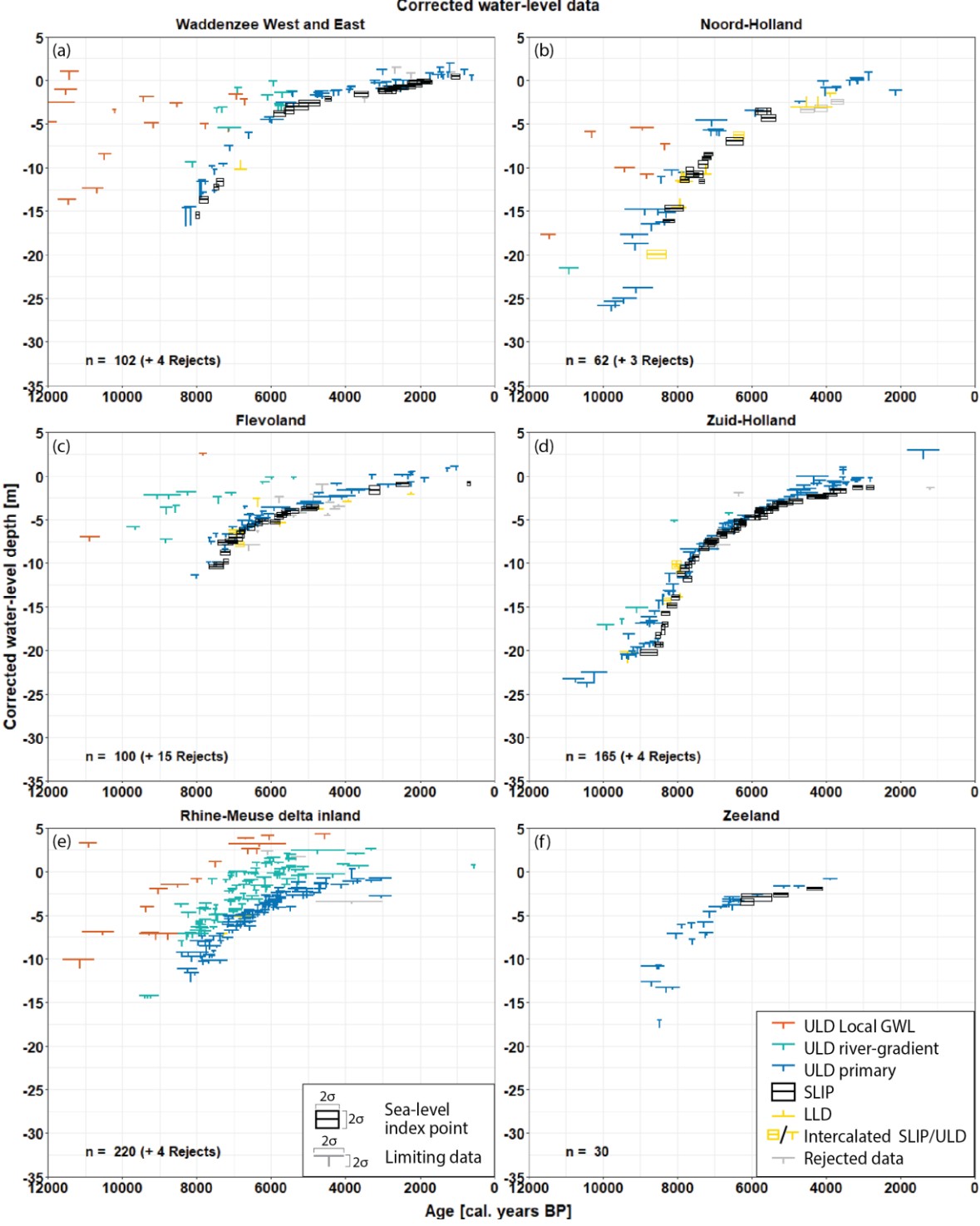

**Figure 9 Age-depth plots of the corrected water level (GWL) for geological indicators per subregion from 12 to 0 ky cal. BP**

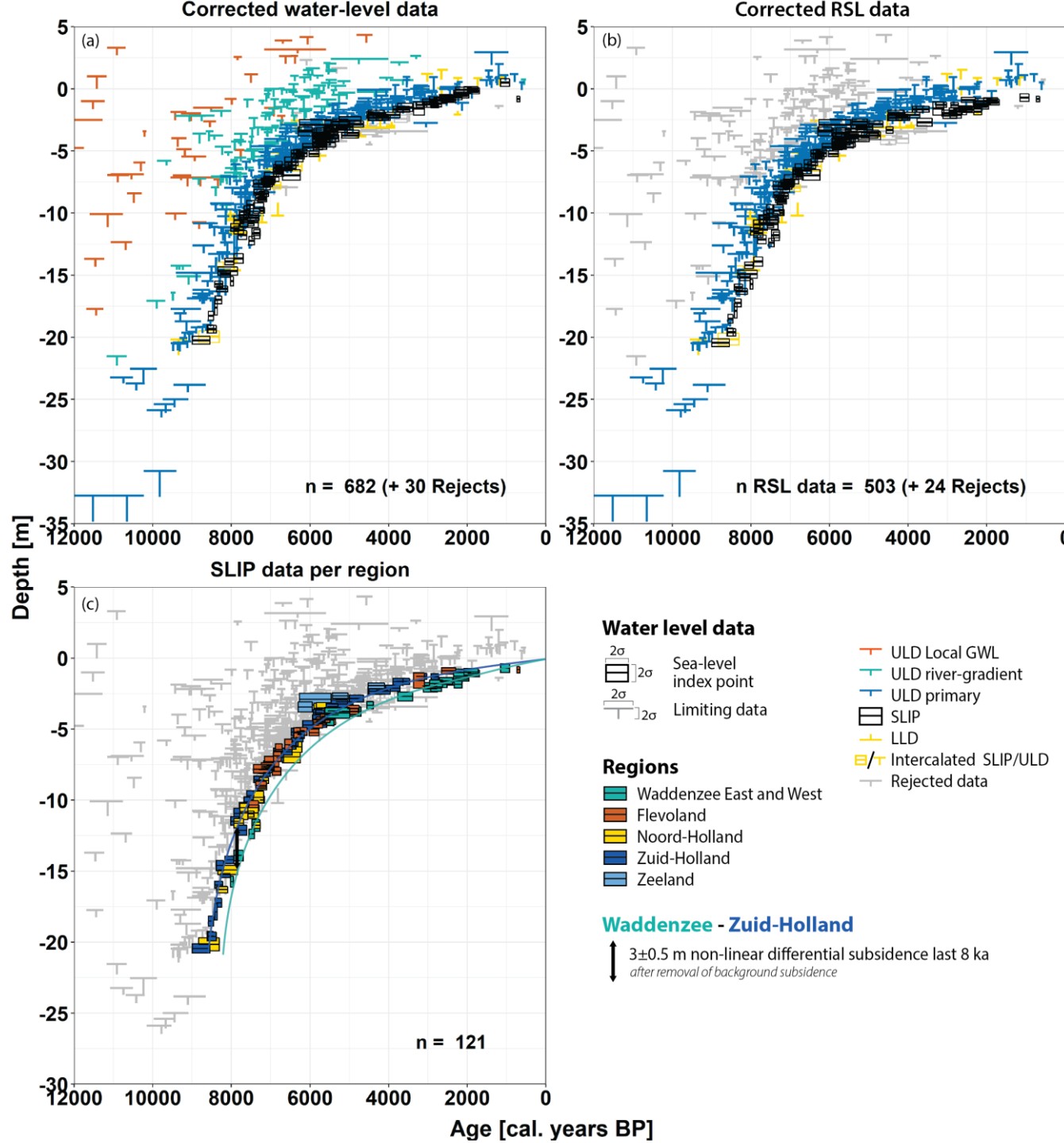

**Figure 10 Age of indicator data plotted against various sample elevations based on different vertical adjustments applied: (a) Using the indicative water-level, and corrected for decompaction, background basin subsidence and anthropogenic deep subsidence; (b) As (a), but incorporating a palaeotidal correction for SLIPs and removing secondary ULD and rejected samples (both in grey); (c)**

**Fully processed SLIP data (all vertical corrections considered), coloured by Region, with tentative curves for the Waddenzee (light blue) and Zuid-Holland SLIP data (dark blue) to illustrate deviation back in time.**

## 6. Discussion

Bringing together water level indicator data shows the prospect of reconstructing relative water level rise on a larger spatial scale. Previously, water level or sea-level reconstructions were constrained locally to areas where there was a straightforward sampling opportunity (i.e. Zuid-Holland: 'Rotterdam', 'Lower RM-delta') and as this area was revisited several times, it resulted in a high data density (e.g. Jelgersma, 1961; Berendsen et al., 2007; Van de Plassche et al., 2010; Hijma and Cohen, 2019). Compilations for other regions were more incidentally executed (Jelgersma, 1961; Kiden et al., 2002; Makaske et al., 2003; Meijles et al., 2018). The newly compiled data set and explicit vertical correction bookkeeping, first of all, create an opportunity to cross-validate the quality and accuracy of the dataset (Sect. 6.1). Furthermore, it allows for superregional water level reconstructions in previously data sparse regions.

### 6.1 Uncertainties and Limitations

In this paper and in the accompanying HOLSEA-NL data set, we attempted to document all Holocene (coastal) water level indicators for the Netherlands that are relevant for reconstructing relative groundwater rise, sea-level rise, and regional subsidence. Given the long history of water level and sea-level research in the Netherlands, a broad range of documentation of suitable data existed, not always easily accessible, from which in the future new details may emerge that could lead to updating/re-processing of individual entries. Furthermore, ongoing and future research is expected to generate further new water level indicator data and gradually increase the HOLSEA-NL data density. Therefore, the compiled data set of water level indicators and their accompanying metadata should be viewed as a living one, deserving to receive updates once every other year or so.

Underlying the dataset, a broad range of documentation on water-level data exists on account of the large diversity of studies for which this data was collected. For some fields in the workbook, the documentation is consistent for all samples, such as information on the geographic location of the sample, the sample age and information on the stratigraphical position of the sample (HOLSEA datasheet Section C. "Fields related to horizontal position of RSL" and Section D.1). For many other fields, it is much harder to be fully consistent, for example where uncertainties in-depth and absolute elevation of the sample were considered. In some original studies, this information was thoroughly documented, while in others, it was not included. Therefore, the uncertainties in sample depth have been re-calculated using the standard calculations provided by the HOLSEA workbook format. The fields related to the depth-related uncertainties (Section D.2 and D.3 in the HOLSEA workbook) and uncertainties in absolute elevation have partially been added.

The second part of the HOLSEA datasheet contains almost exclusively columns marked as interpretation columns (Section D.4 to D.7 in the HOLSEA workbook). This is the section where the additions of the dataset reported in this paper are documented. To align the documentation of this data, the metadata was iteratively reviewed. Specifically, with regard to the interpretation of the samples as either GWL, ULD or SLIP, an effort was made to expand the dataset. The quality of the data depends strongly on the sampling method, sampling depth and what material was used for dating (e.g., bulk or macrofossils). The question arises if sampled material always represents the actual drowning surface. To prevent such ambiguity where possible, we recommend using a systematic approach for future sampling and dating of basal peat layers, in line with what is proposed by Quik et al. (2022; developed for relatively inland peats).

In few original publications, information on the tidal regime near the sampling site is included. It is understandable that this has not been standard practice, since not all samples were originally collected for water-level and sea-level reconstruction purposes. The HOLSEA-NL data base is set up to facilitate incorporating tidal information, and for all SLIP and primary ULD samples, this information is now provided. In addition to the current MHW level, an effort was made to provide the palaeo-tidal MHW level for these samples as well (as described in Sect. 4.1.2). Especially the FBE note fields may be used in future expansions of the database with e.g. Late Holocene archaeological data along former tidal inlets, where appropriate specification is required (e.g. Behre, 2007; Baeteman et al., 2011) (e.g. Behre, 2007; Baeteman et al., 2011). Similarly, the database accommodates specification of basin subsidence and human-induced subsidence corrections - with respective values taken in from external data products for this (as described in Sect. 4.1.3-4). This provides transparent documentation on vertical corrections, their uncertainties, and the optionality of using them, enabling users to assess vertical corrections individually - which was one of the aims of this paper. Additionally, the transparent documentation of the different steps helps interpret the data and its uncertainties.

## 6.2 Data usage

Providing an overview of water-level data in the Netherlands with transparent documentation on the variety of adjustments needed and optional to transform raw data into sea-level indicators, was the main aim of the paper, fulfilled by publishing (De Wit and Cohen, 2024) and documenting (this paper) the HOLSEA-NL database. In this final section, some foreseen usage of the database will be briefly discussed.

First, the data set is intended to be used for relative sea-level reconstruction. For this, the SLIPs, ULD and LLD are relevant input as well as the different tidal corrections (palaeo-tidal or using current MHW). Moreover, the increase in spatial and temporal coverage of the sea-level data, makes it possible to study patterns in the sea-level rise, such as tidal dampening (FBE) (Van de Plassche, 1995a) and the river gradient effect (Louwe Kooijmans, 1972). Especially in the northern parts of the study area, there is now more regional data, from which a start can be made to constrain the timing and extent of FBE sub-regionally (as advocated for in Vis et al., 2015).

A second main application is to use the relative water-level rise documented through the data set, for reconstructing regional scale subsidence patterns. The fields related to the background subsidence correction help remove the effect of the sinking of the North Sea basin from the RSLR signal. This allows for the production of age-depth plots that are directly comparable to regional GIA modelling output. As mentioned and illustrated in Sect. 5.2, spatial intercomparison of rising trends in the data has revealed overprints of differential subsidence (Figure 9 and Figure 10). Over the Holocene, these are attributed to GIA

mainly (2–4 m difference; Sect. 5.2), with basin subsidence an additional minor component, especially in the W and NW sectors of the study area (0.4–1 m extra offset; Sect. 4.1.3). Many earlier sea-level data and GIA modelling combining studies stand as examples of the importance of comparing geological data with numerical GIA modelling output, to verify and constrain the modelling insights (e.g., Shennan and Horton, 2002; Vink et al., 2007; Bradley et al., 2011). Vice versa, modelling forecasts depths of sea-level indicators of a given age in data-scarce regions.

With both data coverage and GIA modelling spatiotemporal resolutions increasing, geological data and modelling-derived insights should be expected to slowly converge. The increase in sea-level data density demonstrated by the HOLSEA-NL dataset (this paper), and specifically the better coverage and more uniform assessment of data from the northern half of the Netherlands, will provide more input to constrain GIA modelling output on a spatial scale. Conversely, a better understanding of the GIA signal in the Netherlands will aid in untangling the Holocene relative sea-level rise signal into constituent

components.

Apart from regional scale sea-level and subsidence reconstructions, the data set can also be used for reconstructing groundwater levels sub-regionally: in inland areas of the coastal plain, in the Rhine-Meuse delta sector, and throughout the coastal plain. In that case, all indicator types provide input, and not the RSL but the GWL (i.e., without tidal correction) is used. Reconstructing

local groundwater levels is the more direct approach when 3D mapping of the build-up of the Holocene wedge is the application, and for studying spatial patterns in the relative water level rise (similar to Cohen, 2005; Koster et al., 2017). This is particularly true for the Rhine-Meuse delta, where previous work on water-level indicator data has shown water-level isochrones to display a downstream gradient (Louwe Kooijmans, 1972; Van Dijk et al., 1991). As a first step, we have explicitly labelled the data points to which this applies (river gradient ULDs). In Sect. 5, this was presented with a focus on the deselection

of such data points when exploring the dataset for spatial patterns and differences in MHW and MSL. Applications that explicitly include this data and that investigate and analyze variability in groundwater table elevations in space and time in the delta flood basins may also be envisaged (e.g., Van Asselen et al., 2017). In future investigations, it might be possible to further analyze the river gradient effect and potentially correct inland water-level indicators for this process, for instance to extend differential subsidence analysis inland. Even without explicit correction, groundwater level isochrones of Rhine-Meuse delta

flood basin peats have been used for analysing local vertical displacement of deposits. For example, to quantify fault offsets across the Peel Boundary Fault zone (Cohen, 2005) and to quantify the degree of compaction-lowering of intercalated peats at shorter and further distances of burying deltaic river branches (Van Asselen, 2011). Furthermore, these water level isochrones

based on basal peat data have provided context for many archaeological excavating studies on the Mesolithic and Neolithic of the Rhine-Meuse delta (e.g., Louwe Kooijmans, 1972; Van der Woude, 1983; Verbruggen, 1992) as well as in the further coastal plain (e.g., Peeters, 2007; Van den Biggelaar et al., 2015).

## 7. Conclusion

This paper presents a compilation of Holocene coastal water-level indicator data from the Netherlands, brought together in a consistent format, using the HOLSEA workbook template. The workbook was expanded and complemented, and processing protocols were adapted to accommodate information on more inland water-level data as well as to make the compilation suitable for reconstructing relative groundwater rise, sea-level rise and regional subsidence. The compilation processed legacy data, as well as more recently produced data, in the majority (>600 out of 712 points) not earlier processed with HOLSEA protocols (104 data points from Zuid Holland being the exception).

Careful and systematic incorporation of sample properties from extensive scattered documentation on individual samples from more than 110 source papers, reports, and a considerable amount of specialised literature (Table 2) on GWL and MSL indicative meaning of peat data from the Dutch setting, allowed for consistent treatment and specification of different depth adjustment possibilities for each sample. The classification of the data in SLIPs, ULDs and LLDs, especially when combined with the locational information and subregion-labelling, should help guide data users in making the right sub-selections for their application.

In conclusion, this paper and the versatile structure of the new HOLSEA-NL data set make the water-level data suitable for multiple usages. On top of that, its open accessibility and documentation make future expansion of the data set possible. In northeastern, northwestern, and southwestern parts of the Netherlands, there are still considerable gaps that would welcome improved coverage with Holocene water level markers, also for cross-validating the current data. Overall, the open access data set can provide input and context for future Holocene water-level and sea-level research, bridging between the large amount of legacy data and newly collected indicator data and unifying these data in a consistent format.

**Table 2 Primary data references, including the number of samples per reference and regions covered, split by source type**

| Reference (primary) | Nr of samples | Regions |
|---|---|---|
| *Source type: Scientific publications* | *562* | |
| Berendsen (1982) | 10 | Rhine-Meuse delta inland |
| Berendsen & Stouthamer (2001) | 45 | Rhine-Meuse delta inland, Zuid-Holland, Flevoland, Zeeland |
| Berendsen et al. (2007) | 26 | Rhine-Meuse delta inland, Zuid-Holland |

| | | |
|---|---|---|
| Busschers et al. (2007) | 1 | Zuid-Holland |
| Candel et al. (2017); Makaske & Maas (2023) | 6 | Waddenzee East |
| Cohen (2003); Cohen (2005); Cohen et al. (2005) | 39 | Rhine-Meuse delta inland |
| De Wit et al. (2024) (This paper; 28 newly dated, 10 legacy data points) | 38 | Rhine-Meuse inland delta, Waddenzee East, Noord-Holland, Flevoland, Zuid-Holland |
| Gotjé (1993), *including Roeleveld & Gotjé (1993)* | 20 | Flevoland |
| Gotjé (1997a) | 3 | Flevoland |
| Gotjé (1997b) | 2 | Flevoland |
| Gouw (2008); Gouw & Erkens (2007) | 9 | Rhine-Meuse delta inland |
| Griede (1978) | 5 | Waddenzee West, Waddenzee East |
| Hijma & Cohen (2010) | 4 | Zuid-Holland |
| Hijma & Cohen (2019) | 4 | Zuid-Holland |
| Hijma et al. (2009); Hijma (2009) | 20 | Rhine-Meuse delta inland, Zuid-Holland |
| Hofstede et al. (1989) | 1 | Rhine-Meuse delta inland |
| Jelgersma (1960) | 2 | Waddenzee East |
| Jelgersma (1961) | 52 | Waddenzee West, Waddenzee East, Noord-Holland, Rhine-Meuse delta inland, Zuid-Holland, Zeeland |
| Jelgersma et al. (1970) | 4 | Noord-Holland |
| Kiden & Vos (2012) | 4 | Waddenzee East |
| Kiden (1989) | 8 | Zeeland (incl. Belgian lower Scheldt) |
| Kiden (1995) | 3 | Zeeland (incl. Belgian lower Scheldt) |
| Kooistra et al. (2006) | 1 | Flevoland |
| Koster et al. (2017) | 6 | Noord-Holland, Zuid-Holland |
| Louwe Kooijmans (1972) | 1 | Flevoland |
| Makaske et al. (2002); Makaske et al. (2003) | 16 | Flevoland |
| Meijles et al. (2018) | 15 | Waddenzee West, Waddenzee East |
| Slupik et al. (2013) | 1 | Zeeland |
| Törnqvist (1993) | 9 | Rhine-Meuse inland delta |
| Törnqvist et al. (1998) | 6 | Rhine-Meuse inland delta |
| Van Asselen (2010) | 6 | Rhine-Meuse inland delta |
| Van Asselen et al. (2017) | 5 | Rhine-Meuse inland delta |
| Van de Meene (1994) | 1 | Noord-Holland |

| | | |
|---|---|---|
| Van de Plassche (1980) | 2 | Rhine-Meuse inland delta |
| Van de Plassche (1982) | 75 | Rhine-Meuse inland delta, Zuid-Holland |
| Van de Plassche (1995b) | 4 | Zuid-Holland |
| Van de Plassche et al. (2010) | 27 | Rhine-Meuse inland delta, Zuid-Holland |
| Van der Spek (1994) | 2 | Waddenzee West |
| Van der Woude (1981) | 3 | Rhine-Meuse inland delta |
| Van Dijk et al. (1991) | 28 | Rhine-Meuse inland delta |
| Van Dinter et al. (2017) | 1 | Rhine-Meuse inland delta |
| Van Heteren et al. (2002) | 3 | Zuid-Holland |
| Van Straaten (1954); Van Straaten and De Jong (1957); Bennema (1954) | 7 | Noord-Holland |
| Verbruggen (1992) | 3 | Rhine-Meuse inland delta |
| Vos et al. (2015), including: *Vos et al. (2011); Vos (2013); Vos & Cohen (2014)* | 14 | Zuid-Holland |
| Vos & Nieuwhof (2021), including: *Vos (1999); Vos & Gerrets (2005); Schrijer et al. (2006); Nieuwhof & Vos (2006); Vos & Van Zijverden (2008); Vos & Waldus (2012); Vos (2015a); Vos & Varwijk (2017); Nicolay et al. (2018); Varwijk & De Langen (2018)* | 12 | Waddenzee West |
| Morzadec-Kerfourn & Delibrias (1972); Delibrias et al. (1974); Ward et al. (2006) | 3 | Southern Bight (Dover transgression path) |
| Weerts & Berendsen (1995) | 1 | Rhine-Meuse inland delta |
| Woldring et al. (2005) | 3 | Waddenzee East |
| Zagwijn (1961) | 1 | Noord-Holland |
| ***Archaeological professional reports*** | ***100*** | |
| Aalbersberg (2018) | 6 | Waddenzee East |
| Bakker (1992) | 1 | Waddenzee East |
| Bouman & Bos (2012) in Hamburg et al. (2012) | 7 | Flevoland |
| Brijker & Van Zijverden (2009) | 3 | Flevoland |
| Brinkkemper et al. (2006) | 23 | Waddenzee East |
| Bulten et al. (2013) | 1 | Zuid-Holland |
| De Moor et al. (2009) | 12 | Flevoland |
| De Moor et al. (2013) | 2 | Flevoland |
| Groenendijk (1997) | 2 | Waddenzee East |
| Hamburg & Knippenberg (2006) | 2 | Flevoland |

| | | |
|---|---|---|
| Kooistra (2012) in Hamburg et al. (2012) | 1 | Flevoland |
| Lohof & Alders (2008) | 3 | Flevoland |
| Lohof et al. (2011) | 13 | Flevoland |
| Osinga & Hekman (2011) | 4 | Flevoland |
| Spek et al. (1997) | 4 | Flevoland |
| Teunissen (1988) | 1 | Rhine-Meuse inland delta |
| Teunissen (1990) | 2 | Rhine-Meuse inland delta |
| Van der Linden (2010a) | 1 | Flevoland |
| Van der Linden (2008) in Van Lil (2008) | 2 | Flevoland |
| Van der Linde (2010b) | 3 | Zuid-Holland |
| Van Dinter (2018) | 3 | Flevoland |
| Van Smeerdijk (2003) | 1 | Flevoland |
| Van Smeerdijk (2004) | 1 | Noord-Holland |
| Van Smeerdijk (2006) | 1 | Flevoland |
| Vos et al. (2008) | 1 | Waddenzee East |
| ***Geological professional reports*** | ***50*** | |
| Barckhausen (1984) | 1 | Waddenzee East (German side of Ems) |
| Bosch & Kok (1994) | 3 | Rhine-Meuse delta inland |
| Cohen et al. (2012) [database publication] | 10 | Rhine-Meuse delta inland, Zuid-Holland, Flevoland |
| De Groot et al. (1996) | 4 | Waddenzee East |
| De Jong (1984) | 6 | Waddenzee West |
| De Jong (1986) | 1 | Noord-Holland |
| De Jong (1989) | 2 | Noord-Holland |
| De Jong (1990) | 1 | Rhine-Meuse delta inland |
| De Jong (1992a) | 1 | Noord-Holland |
| De Jong (1992b) | 1 | Noord-Holland |
| De Mulder & Bosch (1982), including: *Du Burck* (1960, 1972)*; De Jong & Van Regteren Altena* (1972)*; Ente et al.* (1975) | 8 | Noord-Holland |
| De Jong (1988) | 2 | Noord-Holland |
| Verbraeck (1984) | 1 | Rhine-Meuse inland delta |
| Veldkamp (1996) | 1 | Zuid-Holland |
| Vos (1992) | 7 | Zeeland |
| Zagwijn & De Jong (1982) | 1 | Noord-Holland |

## Data availability

The HOLSEA-NL database (https://zenodo.org/doi/10.5281/zenodo.11098446; De Wit and Cohen (2024)), as a scientific product, is open-access available (CC-BY). The data set was compiled by the authors, and contains abundant referencing to a great variety of source publications (part scientific, and referred to in this paper too, part applied regional reports and/or from institutional databases, referencing not doubled in this paper). The HOLSEA-NL database format is compliant with field descriptions in https://www.holsea.org/archive-your-data: see Sect. 4 for added fields specific to our compiling and the Dutch geological setting.

**Team list**. Kim de Wit, Kim M. Cohen, Roderik S. W. van de Wal

**Author contribution.** KW and KMC reviewed literature related to data inventory and curated the data set. KW prepared figures, tables and original draft. All authors edited the manuscript and contributed to conceptualization of the manuscript and designing key figures.

**Competing interests.** The authors declare that they have no conflict of interest.

**Acknowledgements.** The research presented in this paper is part of the project Living on Soft Soils: Subsidence and Society (grantnr.: NWA.1160.18.259). This project is funded by the Dutch Research Council (NWO-NWA-ORC), Utrecht University, Wageningen University, Delft University of Technology, Ministry of Infrastructure & Water Management, Ministry of the Interior & Kingdom Relations, Deltares, Wageningen Environmental Research, TNO-Geological Survey of The Netherlands, STOWA, Water Authority: Hoogheemraadschap de Stichtse Rijnlanden, Water Authority: Drents Overijsselse Delta, Province of Utrecht, Province of Zuid-Holland, Municipality of Gouda, Platform Soft Soil, Sweco, Tauw BV, NAM.

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

De Jong, J.: RGD Pal. Lab. Rap. 1009 -Veldgegevens van ontsluitingen bij Heiloo en daarmee samenhangende resultaten van C14-ouderdomsbepalingen, Rijks Geologische Dienst, Haarlem, 1986.

De Jong, J.: RGD Pal. Lab. Rap. 895, 895ab - Gegevens inzake te verrichten C14-ouderdomsbepaingen i.v.m. de genese van de Utrechtse Vecht en de Kromme Amstel, Rijks Geologische Dienst, Haarlem, 1988.

De Jong, J.: RGD Pal. Lab. Rap. 1067 - Pollenanalytisch en C14-onderzoek aan strandvlakteveen uit het stadsdeel Schalkwijk onder Haarlem, Rijks Geologische Dienst, Haarlem, 1989.

De Jong, J.: RGD  Pal. Lab. Rap. 974a - Uitkomst van een C14-bepaling aan veen afkomstig uit een bouwput aan de Hambakenweg te Den Bosch, Rijks Geologische Dienst, Haarlem, 1990.

De Jong, J.: RGD Pal. Lab. Rap. 1161 - Schoorl Strand; veldwaarnemingen en resultaten van pollenanalytisch- en C14-onderzoek, Rijks Geologische Dienst, Haarlem, 1992a.

De Jong, J.: RGD Pal. Lab. Rap. 1178 - Pollenanalytisch- en C14-onderzoek van een boring uit de strandvlakte te Ruigenhoek, Rijks Geologische Dienst, Haarlem, 1992b.

De Jong, J. and van Regteren Altena, J.: Enkele geologische en archeologische waarnemingen in Alkmaars oude stad - Alkmaar, van boederij tot middeleeuwse stad; Alkmaarse Studieën 1, 1972.

De Moor, J. J. W., Bos, J. A. A., Bouman, M., Moolhuizen, C., Exaltus, R. P., Maartense, F. P. A., and Van der Linden, T. J. M.: Definitief Archeologisch Onderzoek in het tracé van de Hanzelijn in het Nieuwe Land, Een interdisciplinaire geo-archeologische waardering van het begraven landschap van Oostelijk Flevoland, 2009.

De Moor, J. J. W., Maurer, A. M., Houchin, R., and Fritzsch, D.: Almere Poort, Godendreef Verstoringsonderzoek 4J4K_I De Distel, https://doi.org/10.17026/dans-zpt-4yfk, 2013.

De Mulder, E. F. J. and Bosch, J. H. A.: Holocene stratigraphy, radiocarbon datings and paleogeography of central and northern North-Holland (The Netherlands)., Mededelingen - Rijks Geologische Dienst, 36, 111–160, 1982.

De Wit, K. and Cohen, K. M.: HOLSEA-NL: Holocene water level and sea-level indicator dataset for the Netherlands (v1.0), https://doi.org/10.5281/zenodo.11098447, 2024.

Delibrias, G., Guillier, M. T., and Labeyrie, J.: Gif Natural Radiocarbon Measurements VIII, Radiocarbon, 16, 15–94, https://doi.org/10.1017/S0033822200001417, 1974.

Du Burck, P.: Enige beschouwingen over het ontstaan van de oudere zeeklei-afzettingen en het oppervlakte-veen in het noordelijk deel van Noord-Holland: uitgaande van stratigrafische gegevens en met behulp van enkele 14C bepalingen en palynologische analysen, STIBOKA, 1960.

Du Burck, P.: De bodemgesteldheid van de Anna Paulownapolder en van de polder Het Koegras, STIBOKA, 1972.

Ente, P. J., Zagwijn, W. H., and Mook, W. G.: The Calais deposits in the vicinity of wieringen and the geogenisis of Northern North Holland, Geologie en Mijnbouw, 54, 1–14, 1975.

Erkens, G., Van der Meulen, M. J., and Middelkoop, H.: Double trouble: subsidence and CO2 respiration due to 1,000 years of Dutch coastal peatlands cultivation, Hydrogeol J, 24, 551–568, https://doi.org/10.1007/s10040-016-1380-4, 2016.

Gotjé, W.: De Holocene laagveenontwikkeling in de randzone van de Nederlandse kustvlakte (Noordoostpolder), Vrije Universiteit, Amsterdam, 1993.

Gotjé, W.: BIAXiaal 40 Het landschap in Zuidelijk Flevoland tussen 9500 en 4300 BP. Een landschapsreconstructie in het gebied Wet Bodembescherming., 1997a.

Gotjé, W.: De vegetatie op en rond een Mesolithische en Vroeg Neolithische vindplaats, Een ecologisch onderzoek aan drie kernen op de vindplaats Hoge Vaart, BIAXiaal, 36, 1997b.

Gouw, M. J. p.: Alluvial architecture of the Holocene Rhine–Meuse delta (the Netherlands), Sedimentology, 55, 1487–1516, https://doi.org/10.1111/j.1365-3091.2008.00954.x, 2008.

Gouw, M. J. P. and Erkens, G.: Architecture of the Holocene Rhine-Meuse delta (the Netherlands) - A result of changing external controls, Netherlands Journal of Geosciences, 86, 23–54, https://doi.org/10.1017/S0016774600021302, 2007.

Griede, J. W.: Het ontstaan van Frieslands Noordhoek., Universiteit Amsterdam, 186 pp., 1978.

Groenendijk, H., Mook-Kamps, E., and Elerie, J. N. H.: Op zoek naar de horizon: het landschap van Oost-Groningen en zijn bewoners tussen 8000 voor Chr. en 1000 na Chr, REGIO-PRojekt Uitgevers Groningen, 1997.

Hamburg, T. and Knippenberg, S.: Proefsleuven op drie locaties binnen het tracé van de Hanzelijn 'Oude Land,' 2006.

Hamburg, T., Müller, A., and Quadflieg, B.: Dronten Swifterbant N23 vindplaats 5 Opgraving (V1), https://doi.org/10.17026/dans-zeh-g9g8, 2012.

Heaton, T. J., Köhler, P., Butzin, M., Bard, E., Reimer, R. W., Austin, W. E. N., Ramsey, C. B., Grootes, P. M., Hughen, K. A., Kromer, B., Reimer, P. J., Adkins, J., Burke, A., Cook, M. S., Olsen, J., and Skinner, L. C.: Marine20—The Marine Radiocarbon Age Calibration Curve (0–55,000 cal BP), Radiocarbon, 62, 779–820, https://doi.org/10.1017/RDC.2020.68, 2020.

Hijma, M. P.: From river valley to estuary : the early-mid Holocene transgression of the Rhine-Meuse valley, The Netherlands, Dissertation, Utrecht University, Royal Dutch Geographical Society, 2009.

Hijma, M. P. and Cohen, K. M.: Timing and magnitude of the sea-level jump preluding the 8200 yr event, Geology, 38, 275–278, https://doi.org/10.1130/G30439.1, 2010.

Hijma, M. P. and Cohen, K. M.: Holocene sea-level database for the Rhine-Meuse Delta, The Netherlands: Implications for
the pre-8.2 ka sea-level jump, Quaternary Science Reviews, 214, 68–86, https://doi.org/10.1016/j.quascirev.2019.05.001, 2019.

Hijma, M. P., Cohen, K. M., Hoffmann, G., Spek, A. J. F. V. der, and Stouthamer, E.: From river valley to estuary: the evolution of the Rhine mouth in the early to middle Holocene (western Netherlands, Rhine-Meuse delta), Netherlands Journal of Geosciences, 88, 13–53, https://doi.org/10.1017/S0016774600000986, 2009.

Hijma, M. P., Engelhart, S. E., Törnqvist, T. E., Horton, B. P., Hu, P., and Hill, D. F.: A protocol for a geological sea-level database, in: Handbook of Sea-Level Research, John Wiley & Sons, Ltd, 536–553, https://doi.org/10.1002/9781118452547.ch34, 2015.

Hoek, W. Z.: Late-glacial and early Holocene climatic events and chronology of vegetation development in the Netherlands, Veget Hist Archaebot, 6, 197–213, https://doi.org/10.1007/BF01370442, 1997.

Hofstede, J., Berendsen, H., and Janssen, C.: Holocene palaeogeography and palaeoecology of the fluvial area near Maurik (Neder-Betuwe, The Netherlands), Geologie en Mijnbouw, 68, 409–419, 1989.

Jelgersma, S.: Die Palynologische und C14-Untersuchung einiger Torfprofile aus dem NS-Profil Meedhuizen-Farmsum, Das Ems-Estuarium (Nordsee). Verh. K. Ned. Geol.-Mijnb. k. Gen. Geol. Ser, 19, 25–32, 1960.

Jelgersma, S.: Holocene sea-level changes in the Netherlands, Ph. D. dissertation, Leiden University, Leiden, 1961.

Jelgersma, S., de Jong, S., Zagwijn, W., and van Regteren Altena, J.: The coastal dunes of the western Netherlands: geology, vegetational history and archeology, Rijks Geologische Dienst, 1970.

Kasse, C. and Aalbersberg, G.: A complete Late Weichselian and Holocene record of aeolian coversands, drift sands and soils forced by climate change and human impact, Ossendrecht, the Netherlands, Netherlands Journal of Geosciences, 98, e4, https://doi.org/10.1017/njg.2019.3, 2019.

Khan, N. S., Horton, B. P., Engelhart, S., Rovere, A., Vacchi, M., Ashe, E. L., Törnqvist, T. E., Dutton, A., Hijma, M. P., and Shennan, I.: Inception of a global atlas of sea levels since the Last Glacial Maximum, Quaternary Science Reviews, 220, 359–371, https://doi.org/10.1016/j.quascirev.2019.07.016, 2019.

Kiden, P.: The Late Holocene evolution of the perimarine part of the River Scheldt, The Quaternary and Tertiary Geology of the Southern Bight, North Sea. Brussel: Belgische Geologische Dienst, 173–184, 1989.

Kiden, P.: Holocene relative sea-level change and crustal movement in the southwestern Netherlands, Marine Geology, 124, 21–41, https://doi.org/10.1016/0025-3227(95)00030-3, 1995.

Kiden, P. and Vos, P. C.: Holocene relative sea-level change and land movements in the northern Netherlands–a first assessment, in: 3rd IGCP588-Conference 'Preparing for Coastal Change'Conference Program–Book of Abstracts. Christian-Albrechts-Universität zu Kiel (Kiel), 2012.

Kiden, P., Denys, L., and Johnston, P.: Late Quaternary sea-level change and isostatic and tectonic land movements along the Belgian-Dutch North Sea coast: geological data and model results, J. Quaternary Sci., 17, 535–546, https://doi.org/10.1002/jqs.709, 2002.

Kiden, P., Makaske, B., and Van de Plassche, O.: Waarom verschillen de zeespiegelreconstructies voor Nederland?, Grondboor & Hamer, 62, 54–61, 2008.

Kooi, H., Johnston, P., Lambeck, K., Smither, C., and Ronald Molendijk: Geological causes of recent (∼100 yr) vertical land movement in the Netherlands, Tectonophysics, 299, 297–316, https://doi.org/10.1016/S0040-1951(98)00209-1, 1998.

Kooistra, M. J., Kooistra, L. I., Van Rijn, P., and Sass-Klaassen, U.: Woodlands of the past—The excavation of wetland woods at Zwolle-Stadshagen (the Netherlands): Reconstruction of the wetland wood in its environmental context, Netherlands journal of geosciences, 85, 37–60, 2006.

Koster, K., Stafleu, J., and Cohen, K. M.: Generic 3D interpolation of Holocene base-level rise and provision of accommodation space, developed for the Netherlands coastal plain and infilled palaeovalleys, Basin Res, 29, 775–797, https://doi.org/10.1111/bre.12202, 2017.

Lambeck, K.: Late Devensian and Holocene shorelines of the British Isles and North Sea from models of glacio-hydro-isostatic rebound, JGS, 152, 437–448, https://doi.org/10.1144/gsjgs.152.3.0437, 1995.

Lohof, E. and Alders, P. G.: Hattemerbroek Bedrijventerrein Noord (2), https://doi.org/10.17026/dans-zyt-zzer, 2008.

Lohof, E., Hamburg, T., and Flamman, J.: Steentijd opgespoord. Archeologisch onderzoek in het tracé van de Hanzelijn-Oude Land, Archol & ADC-Archeoprojecten, Leiden & Amersfoort (Archol report 138 & ADC report 2576), 2011.

Louwe Kooijmans, L.: The Rhine/Meuse Delta. Four studies on its prehistoric occupation and Holocene geology, Leiden University, Leiden, 1972.

Makaske, B. and Maas, G. J.: Different hydrological controls causing variable rates of Holocene peat growth in a lowland valley system, north-eastern Netherlands; implications for valley peatland restoration, The Holocene, 33, 960–974, https://doi.org/10.1177/09596836231169985, 2023.

Makaske, B., Smeerdijk, D. G. van, Mulder, J. R., and Spek, T.: De stijging van de waterspiegel nabij Almere in de periode 5300-2300 v. Chr, Alterra, Wageningen, 2002.

Makaske, B., Smeerdijk, D. G. V., Peeters, H., Mulder, J. R., and Spek, T.: Relative water-level rise in the Flevo lagoon (The Netherlands), 5300-2000 cal. yr BC: an evaluation of new and existing basal peat time-depth data, Netherlands Journal of Geosciences, 82, 115–131, https://doi.org/10.1017/S0016774600020680, 2003.

Meijles, E. W., Kiden, P., Streurman, H.-J., Van der Plicht, J., Vos, P. C., Gehrels, W. R., and Kopp, R. E.: Holocene relative mean sea-level changes in the Wadden Sea area, northern Netherlands, Journal of Quaternary Science, 33, 905–923, https://doi.org/10.1002/jqs.3068, 2018.

Morzadec-Kerfourn, M. T. and Delibrias, G.: Analyses polliniques et datations radiocarbone des sediments quaternaires preleves en Manche centrale et orientale, Memoires du Bureau de Recherche Geologique et Miniere, 79, 160–5, 1972.

NAM: Aanvraag Instemming Winningsplan Gaag-Monster, 2017.

NAM: Statusrapport 2020 en Prognose tot het jaar 2080, 2020.

Nicolay, J., Schepers, M., Postma, D., and Kaspers, A.: Firdgum: pioniers, boeren en terpbewoners, in: De geschiedenis van terpen-en wierdenland: Een verhaal in ontwikkeling, Vereniging voor Terpenonderzoek, 133–148, 2018.

Nieuwhof, A. and Vos, P. C.: Landschap en bewoningsgeschiedenis. In: Nieuwhof, A. (ed.): De wierde Wierum (provincie Groningen). Een archeologisch steilkantonderzoek (= Groningen Archaeological Studies 3)., Barkhuis Publishing & Groningen University Library (Groningen), 2006.

Osinga, M. and Hekman, J. J.: Archeologisch onderzoek Hanzelijn deelgebieden XIV en XV, Grontmij, Assen, 2011.

Peeters, J., Busschers, F. S., Stouthamer, E., Bosch, J. H. A., Van den Berg, M. W., Wallinga, J., Versendaal, A. J., Bunnik, F. P. M., and Middelkoop, H.: Sedimentary architecture and chronostratigraphy of a late Quaternary incised-valley fill: A case study of the late Middle and Late Pleistocene Rhine system in the Netherlands, Quaternary Science Reviews, 131, 211–236, https://doi.org/10.1016/j.quascirev.2015.10.015, 2016.

Peeters, J. H. M.: Hoge Vaart-A27 in context: towards a model of mesolithic - neolithic land use dynamics as a framework for archaeological heritage management, AmsersfoortRijksdienst voor Archeologie, Cultuurlandschap en Monumenten, 2007.

Pierik, H. J. and Cohen, K. M.: The use of geological, geomorphological and soil mapping products in palaeolandscape reconstructions for the Netherlands, Netherlands Journal of Geosciences, 99, e9, https://doi.org/10.1017/njg.2020.8, 2020.

Pierik, H. J., Cohen, K. M., Vos, P. C., Van der Spek, A. J. F., and Stouthamer, E.: Late Holocene coastal-plain evolution of the Netherlands: the role of natural preconditions in human-induced sea ingressions, Proceedings of the Geologists' Association, 128, 180–197, https://doi.org/10.1016/j.pgeola.2016.12.002, 2017.

Pierik, H. J., Moree, J. I. M., van der Werf, K. M., Roelofs, L., Albernaz, M. B., Wilbers, A., van der Valk, B., van Dinter, M., Hoek, W. Z., de Haas, T., and Kleinhans, M. G.: Vegetation and peat accumulation steer Holocene tidal–fluvial basin filling and overbank sedimentation along the Old Rhine River, The Netherlands, Sedimentology, 70, 179–213, https://doi.org/10.1111/sed.13038, 2023.

Pons, L. J. and Wiggers, A. J.: The holocene wordingsgeschiedenis van Noord-Holland en het Zuiderzeegebied = The holocene genesis of the province of North-Holland and the Zuyder Sea region, 1960.

Quik, C., Van der Velde, Y., Harkema, T., Van der Plicht, H., Quik, J., Van Beek, R., and Wallinga, J.: Using legacy data to reconstruct the past? Rescue, rigour and reuse in peatland geochronology, Earth Surf Processes Landf, 46, 2607–2631, https://doi.org/10.1002/esp.5196, 2021.

Quik, C., Palstra, S. W. L., Van Beek, R., Van der Velde, Y., Candel, J. H. J., Van der Linden, M., Kubiak-Martens, L., Swindles, G. T., Makaske, B., and Wallinga, J.: Dating basal peat: The geochronology of peat initiation revisited, Quaternary Geochronology, 72, 101278, https://doi.org/10.1016/j.quageo.2022.101278, 2022.

Schrijer, E., Lohof, E., and Waldus, W. B.: Tzummarum, rotonde (gem. Franekeradeel). Een archeologische opgraving en een begeleiding., ADC ArcheoProjecten, Amersfoort, 2006.

Shennan, I.: Interpretation of Flandrian sea-level data from the Fenland, England, Proceedings of the Geologists' Association, 93, 53–63, https://doi.org/10.1016/S0016-7878(82)80032-1, 1982.

Shennan, I. and Horton, B.: Holocene land- and sea-level changes in Great Britain, J Quaternary Science, 17, 511–526, https://doi.org/10.1002/jqs.710, 2002.

Shennan, I., Lambeck, K., Flather, R., Horton, B., McArthur, J., Innes, J., Lloyd, J., Rutherford, M., and Wingfield, R.: Modelling western North Sea palaeogeographies and tidal changes during the Holocene, SP, 166, 299–319, https://doi.org/10.1144/GSL.SP.2000.166.01.15, 2000.

Simon, K. M. and Riva, R. E. M.: Uncertainty Estimation in Regional Models of Long-Term GIA Uplift and Sea Level Change: An Overview, Journal of Geophysical Research: Solid Earth, 125, e2019JB018983, https://doi.org/10.1029/2019JB018983, 2020.

Slupik, A. A., Wesselingh, F. P., Mayhew, D. F., Janse, A. C., Dieleman, F. E., Strydonck, M. van, Kiden, P., Burger, A. W., and Reumer, J. W. F.: The role of a proto-Schelde River in the genesis of the southwestern Netherlands, inferred from the Quaternary successions and fossils in Moriaanshoofd Borehole (Zeeland, the Netherlands), Netherlands Journal of Geosciences, 92, 69–86, https://doi.org/10.1017/S0016774600000299, 2013.

Spek, T., Bisdom, E. B. A., and van Smeerdijk, D. G.: Verdronken dekzandgronden in Zuidelijk Flevoland (archeologische opgraving `A27-Hoge Vaart'); een interdisciplinaire studie naar de veranderingen van bodem en landschap in het Mesolithicum en Vroeg-Neolithicum, Staring Centrum, Netherlands, 1997.

Stafleu, J., Maljers, D., Busschers, F., Gunnink, J., Schokker, J., Dambrink, R. M., Hummelman, H. J., and Schijf, M. L.: GeoTop modellering, TNO, 2012.

Steffelbauer, D. B., Riva, R. E. M., Timmermans, J. S., Kwakkel, J. H., and Bakker, M.: Evidence of regional sea-level rise acceleration for the North Sea, Environ. Res. Lett., 17, 074002, https://doi.org/10.1088/1748-9326/ac753a, 2022.

Stouthamer, E. and Berendsen, H. J. A.: Factors Controlling the Holocene Avulsion History of the Rhine-Meuse Delta (The Netherlands), Journal of Sedimentary Research, 70, 1051–1064, https://doi.org/10.1306/033000701051, 2000.

Teunissen, D.: De bewoningsgeschiedenis van Nijmegen en omgeving, haar relatie tot de landschapsbouw en haar weerspiegeling in palynologische gegevens. Mededelingen van de Afdeling Biogeologie van de Sectie Biologie van de Katholieke Universiteit van Nijmegen, 108 pp., 1988.

Teunissen, D.: Palynologisch onderzoek in het oostelijk rivierengebied: een overzicht. Mededelingen van de Afdeling Biogeologie van de Sectie Biologie van de Katholieke Universiteit van Nijmegen 16, 161 pp., 1990.

Törnqvist, T., Weerts, H. J. T., and Berendsen, H.: Definition of two new members in the upper Kreftenheye and Twente Formations (Quaternary, the Netherlands): a final solution to persistent confusion?, Geologie en Mijnbouw, 72, 251–264, 1994.

Törnqvist, T. E.: Holocene alternation of meandering and anastomosing fluvial systems in the Rhine-Meuse delta (central Netherlands) controlled by sea-level rise and subsoil erodibility, Journal of Sedimentary Research, 63, 683–693, 1993.

Törnqvist, T. E., Van Ree, M. H. M., Van 'T Veer, R., and Van Geel, B.: Improving Methodology for High-Resolution
Reconstruction of Sea-Level Rise and Neotectonics by Paleoecological Analysis and AMS [14] C Dating of Basal Peats, Quat. res., 49, 72–85, https://doi.org/10.1006/qres.1997.1938, 1998.

Törnqvist, T. E., González, J. L., Newsom, L. A., Van der Borg, K., De Jong, A. F. M., and Kurnik, C. W.: Deciphering Holocene sea-level history on the U.S. Gulf Coast: A high-resolution record from the Mississippi Delta, GSA Bulletin, 116, 1026–1039, https://doi.org/10.1130/B2525478.1, 2004.

Uehara, K., Scourse, J. D., Horsburgh, K. J., Lambeck, K., and Purcell, A. P.: Tidal evolution of the northwest European shelf seas from the Last Glacial Maximum to the present, Journal of Geophysical Research: Oceans, 111, https://doi.org/10.1029/2006JC003531, 2006.

Van Asselen, S.: Peat compaction in deltas : implications for Holocene delta evolution, Dissertation, Koninklijk Nederlands Aardrijkskundig Genootschap, 2010.

Van Asselen, S.: The contribution of peat compaction to total basin subsidence: implications for the provision of accommodation space in organic-rich deltas: The contribution of peat compaction to basin subsidence, Basin Research, 23, 239–255, https://doi.org/10.1111/j.1365-2117.2010.00482.x, 2011.

Van Asselen, S., Stouthamer, E., and Van Asch, Th. W. J.: Effects of peat compaction on delta evolution: A review on processes, responses, measuring and modeling, Earth-Science Reviews, 92, 35–51,
https://doi.org/10.1016/j.earscirev.2008.11.001, 2009.

Van Asselen, S., Cohen, K. M., and Stouthamer, E.: The impact of avulsion on groundwater level and peat formation in delta floodbasins during the middle-Holocene transgression in the Rhine-Meuse delta, The Netherlands, The Holocene, 27, 1694–1706, https://doi.org/10.1177/0959683617702224, 2017.

Van de Meene, J. W.: The shoreface-connected ridges along the central Dutch coast, 1994.

Van de Plassche, O.: Compaction and Other Sources of Error in Obtaining Sea-Level Data: Some Results and Consequences, E&G Quaternary Science Journal, 30, 171–182, https://doi.org/10.3285/eg.30.1.14, 1980.

Van de Plassche, O.: Sea-level change and water-level movements in the Netherlands during the Holocene, Mededelingen Rijks Geologische Dienst, 36, 1–93, 1982.

Van de Plassche, O.: Sea-level research : a manual for the collection and evaluation of data, Geo Books, Norwich, 1986.

Van de Plassche, O.: Evolution of the intra-coastal tidal range in the Rhine-Meuse delta and Flevo Lagoon, 5700-3000 yrs cal B.C., Marine Geology, 124, 113–128, https://doi.org/10.1016/0025-3227(95)00035-W, 1995a.

Van de Plassche, O.: Periodic clay deposition in a fringing peat swamp in the lower Rhine-Meuse river area, 5,400–3,400 cal BC, Journal of Coastal Research, 95–102, 1995b.

Van de Plassche, O., Bohncke, S. J. P., Makaske, B., and Van der Plicht, J.: Water-level changes in the Flevo area, central
Netherlands (5300–1500 BC): implications for relative mean sea-level rise in the Western Netherlands, Quaternary International, 133–134, 77–93, https://doi.org/10.1016/j.quaint.2004.10.009, 2005.

Van de Plassche, O., Makaske, B., Hoek, W. Z., Konert, M., and Van Der Plicht, J.: Mid-Holocene water-level changes in the lower Rhine-Meuse delta (western Netherlands): implications for the reconstruction of relative mean sea-level rise,

palaeoriver-gradients and coastal evolution, Netherlands Journal of Geosciences, 89, 3–20, https://doi.org/10.1017/S0016774600000780, 2010.

Van den Berg, M. and Beets, D.: Saalian glacial deposits and morphology in The Netherlands, Tills and Glaciotectonics. Balkema, Rotterdam, 235–251, 1987.

Van den Biggelaar, D. F. A. M., Kluiving, S. J., Bohncke, S. J. P., Van Balen, R. T., Kasse, C., Prins, M. A., and Kolen, J.: Landscape potential for the adoption of crop cultivation: Role of local soil properties and groundwater table rise during 6000–5400 BP in Flevoland (central Netherlands), Quaternary International, 367, 77–95, https://doi.org/10.1016/j.quaint.2014.09.063, 2015.

Van den Brenk, S., Huisman, H., Willemse, N. W., Smit, B., and Van Os, B. J. H.: Magnetometer mapping of drowned prehistoric landscapes for Archaeological Heritage Management in the Netherlands, Archaeological Prospection, n/a, https://doi.org/10.1002/arp.1925, 2023.

Van der Linden, M.: Palynologisch onderzoek aan een veen- en kleipakket uit het Laat-Mesolithicum bij Almere-De Vaart, BIAX, Zaandam, 2010a.

Van der Linden, M.: Verlaten donken onder het veen? Paleoecologisch onderzoek aan een veenpakket bij Dinteloord, BIAX, Zaandam, 2010b.

Van der Meulen, M. J., Doornenbal, J. C., Gunnink, J. L., Stafleu, J., Schokker, J., Vernes, R. W., Geer, F. C. van, Gessel, S. F. van, Heteren, S. van, Leeuwen, R. J. W. van, Bakker, M. a. J., Bogaard, P. J. F., Busschers, F. S., Griffioen, J., Gruijters, S. H. L. L., Kiden, P., Schroot, B. M., Simmelink, H. J., Berkel, W. O. van, Krogt, R. A. A. van der, Westerhoff, W. E., and Daalen, T. M. van: 3D geology in a 2D country: perspectives for geological surveying in the Netherlands, Netherlands Journal of Geosciences, 92, 217–241, https://doi.org/10.1017/S0016774600000184, 2013.

Van der Molen, J. and De Swart, H. E.: Holocene tidal conditions and tide-induced sand transport in the southern North Sea, Journal of Geophysical Research: Oceans, 106, 9339–9362, https://doi.org/10.1029/2000JC000488, 2001.

Van der Spek, A. J. F.: Large-scale evolution of Holocene tidal basins in the Netherlands, Universiteit Utrecht Faculteit Aardwetenschappen, 1994.

Van der Woude, J. D.: Holocene paleoenvironmental evolution of a perimarine fluviatile area, Ph. D. dissertation, Vrije Universiteit Amsterdam, 1981.

Van der Woude, J. D.: Holocene paleoenvironmental evolution of a perimarine fluviatile area: geology and paleobotany of the area surrounding the archeological excavation at the Hazendonk river dune (western Netherlands), Leiden U.P. Modderman Stichting., Leiden, 124 pp., 1983.

Van Dijk, G. J.: Holocene water level development in The Netherlands' river area; implications for sea-level reconstruction, Geologie en Mijnbouw, 70, 311–326, 1991.

Van Dijk, G. J., Berendsen, H. J. A., and Roeleveld, W.: Holocene water level development in The Netherlands' river area; implications for sea-level reconstruction, Geologie en Mijnbouw, 70, 311–326, 1991.

Van Dinter, M.: Hoofdstuk 10 Fysische Geografie, p.119-133; in Waldus, W.B., (Ed.) (2018) De opgraving en lichting van de 15e eeuwse IJsselkogge, ADC monografie 24., ADC ArcheoProjecten, 2018.

Van Dinter, M., Cohen, K. M., Hoek, W. Z., Stouthamer, E., Jansma, E., and Middelkoop, H.: Late Holocene lowland fluvial archives and geoarchaeology: Utrecht's case study of Rhine river abandonment under Roman and Medieval settlement, Quaternary Science Reviews, 166, 227–265, https://doi.org/10.1016/j.quascirev.2016.12.003, 2017.

Van Heteren, S., Van der Spek, A., and De Groot, T.: Architecture of a preserved Holocene tidal complex offshore the Rhine-Meuse river mouth, The Netherlands, Netherlands Institute of Applied Geoscience TNO - National Geological Survey, 2002.

Van Lil, R.: Aanleg N23 tussen Lelystad en Dronten, 2008.

Van Smeerdijk, D. G.: Pollenonderzoek aan materiaal uit de top van een Pleistocene dekzandrug in Almere Hout ten behoeve van de Cursus Archeologie, BIAX, Zaandam, 2003.

Van Smeerdijk, D. G.: Palynologisch onderzoek en 14C AMS datering van een venige laag uit de locatie Ripperda-complex in Haarlem, BIAX, Zaandam, 2004.

Van Smeerdijk, D. G.: Palynologisch onderzoek en datering van de overgang van het pleistocene zand naar het afdekkende veen bij de Noorderplassen-West in Almere., BIAX, Zaandam, 2006.

Van Straaten, L. M. J. U.: Radiocarbon datings and changes of sea level at Velzen (Netherlands), Geologie en Mijnbouw (NW. SER.), 16, 247–253, 1954.

Van Straaten, L. M. J. U. and De Jong, J.: The excavation at Velsen. A Detailed Study of Upper-Pleistocene and Holocene Stratigraphy, Verh. Kon. Ned. Geol.-Mijnbouwk. Gen., Geol. Ser., Deel XVII, Tweede Stuk, 93–99, 1957.

Van Veen, J.: Eb-en vloedschaar systemen in de Nederlandse getijwateren, Tijdschrift Koninklijk Nederlands Aardrijkskundig Genootschap, 67, 303–325, 1950.

Van Veen, J., Van der Spek, A. J. F., Stive, M. J. F., and Zitman, T.: Ebb and Flood Channel Systems in the Netherlands Tidal Waters, coas, 2005, 1107–1120, https://doi.org/10.2112/04-0394.1, 2005.

Varwijk, T. and de Langen, G.: Terpzoolopgraving Wommels-Stapert 2014 (GIA 138). Terug na 20 jaar: Nieuw archeologisch onderzoek aan de commercieel afgegraven terp Stapert bij Wommels in het hart van Westergo (Friesland), Groninger Instituut voor Archeologie, Rijksuniversiteit Groningen, 2018.

Veldkamp, M. A.: Pollenanalytische en C14-dateringen van de boring Maassluis 37B/226. Palaeobotanie Kenozoïcum 1255A, Rijks Geologische Dienst, Haarlem, 1996.

Verbraeck, A.: Toelichtingen bij de Geologische kaart van Nederland 1: 50.000, Blad Tiel West (39 W) en Blad Tiel Oost (39 O). Rijks Geologische Dienst (Haarlem), 1984.

Verbruggen, M.: Geoarchaeological prospection of the Rommertsdonk, Analecta Praehistorica Leidensia 25: The end of our third decade: Papers written on the occasion of the 30th anniversary of the Institutte of prehistory, volume I, 25, 119–128, 1992.

Vermeersen, B. L. A., Slangen, A. B. A., Gerkema, T., Baart, F., Cohen, K. M., Dangendorf, S., Duran-Matute, M., Frederikse, T., Grinsted, A., Hijma, M. P., Jevrejeva, S., Kiden, P., Kleinherenbrink, M., Meijles, E. W., Palmer, M. D., Rietbroek, R., Riva, R. E. M., Schulz, E., Slobbe, D. C., Simpson, M. J. R., Sterlini, P., Stocchi, P., Van de Wal, R. S. W., and Van der Wegen, M.: Sea-level change in the Dutch Wadden Sea, Netherlands Journal of Geosciences, 97, 79–127, https://doi.org/10.1017/njg.2018.7, 2018.

Vink, A., Steffen, H., Reinhardt, L., and Kaufmann, G.: Holocene relative sea-level change, isostatic subsidence and the radial viscosity structure of the mantle of northwest Europe (Belgium, the Netherlands, Germany, southern North Sea), Quaternary Science Reviews, 26, 3249–3275, https://doi.org/10.1016/j.quascirev.2007.07.014, 2007.

Vis, G.-J., Cohen, K. M., Westerhoff, W. E., Veen, J. H. T., Hijma, M. P., van der Spek, A. J. F., and Vos, P. C.: Paleogeography, in: Handbook of Sea-Level Research, John Wiley & Sons, Ltd, 514–535, https://doi.org/10.1002/9781118452547.ch33, 2015.

Vogel, J. C. and Waterbolk, H. T.: Groningen Radiocarbon Dates IV, Radiocarbon, 5, 163–202, https://doi.org/10.1017/S0033822200036857, 1963.

Vos, P.: Op veilige afstand van de Marne: geologische en paleolandschappelijke waarnemingen in Achlum. In: Nicolay, J.A.W. & de Langen, G.(eds): Graven aan de voet van de Achlumer dorpsterp. Archeologische sporen rondom een terpnederzetting (= Jaarverslagen van de Vereniging voor Terpenonderzoek 97)., in: Graven aan de voet van de Achlumer dorpsterp. 1310 Archeologische sporen rondom een terpnederzetting (= Jaarverslagen van de Vereniging voor Terpenonderzoek 97)., Vereniging voor Terpenonderzoek, Groningen, 31–47, 2015a.

Vos, P. C.: Toelichting kaartblad 43/49 West en 49 Oost: concept toelichting 43/49 West, Holocene deel, Rijks Geologische Dienst, Distrikt Noord, Haarlem, 1992.

Vos, P. C.: The Subatlantic evolution of the coastal area around the Wijnaldum-Tjitsma terp. With a contribution by B.A.M. 1315 Baardman. In: Besteman, J.C., Bos, J.M., Gerrets, D.A., Heidinga, H.A. & De Koning, J. (eds): The Excavations at Wijnaldum. Reports on Frisia in Roman and Medieval times. Volume I., Balkema (Rotterdam/Brookfield), 33–72 pp., 1999.

Vos, P. C.: Geologisch en paleolandschappelijk onderzoek Yangtzehaven (Maasvlakte, Rotterdam). Deltares report 1206788-000-BGS-000187., Deltares, 2013.

Vos, P. C.: Origin of the Dutch coastal landscape: Long-term landscape evolution of the Netherlands during the Holocene, 1320 described and visualized in national, regional and local palaeogeographical map series, Barkhuis, 373 pp., 2015b.

Vos, P. C. and Cohen, K. M.: Landscape genesis and palaeogeography. In: Moree, J.M., Sier, M.M. (Eds.), Interdisciplinary Archaeological Research Programme Maasvlakte 2, vol. 566., BOOR Rotterdam, 2014.

Vos, P. C. and Gerrets, D. A.: Archaeology: a major tool in the reconstruction of the coastal evolution of Westergo (northern Netherlands), Quaternary International, 133, 61–75, 2005.

Vos, P. C. and Nieuwhof, A.: Late-Holocene sea-level reconstruction (1200 BC–AD 100) in the Westergo terp region of the northern Netherlands, Netherlands Journal of Geosciences, 100, e3, https://doi.org/10.1017/njg.2021.1, 2021.

Vos, P. C. and Van Zijverden, W. K.: Landschappelijke ligging. In: Dijkstra, J. & Nicolay, J.A.W. (eds): Een terp op de schop. Archeologisch onderzoek op het Oldehoofsterkerkhof te Leeuwarden, in: ADC Monografie, vol. 3, 25–42, 2008.

Vos, P. C. and Varwijk, T.: Paleolandschappelijke opname Saksenoord (GIA 133), DANS Data Station Archaeology, 2017.

Vos, P. C. and Waldus, W. B.: Landschap en bewoning: over terpen, kwelderwallen en de bedijkingsgeschiedenis, Middeleeuwse bewoningssporen bij Beetgumermolen (= ADC-rapport 3213). ADC-ArcheoProjecten (Amersfoort), 62–65, 2012.

Vos, P. C., Meijer, T., and Van Os, B.: Bodem en geologie. Paleolandschappelijk onderzoek op en rond De Bloemert, in: Nicolay, J.A.W. (Ed.), Opgravingen bij Midlaren: 5000 jaar wonen tussen Hondsrug en Hunzedal (Deel I)., Barkhuis Publishing & Groningen University Library, Groningen, 17–39 pp., 2008.

Vos, P. C., Bazelmans, J., Weerts, H. J. T., and Van de Meulen, M. J.: Atlas Van Nederland in Het Holoceen., RCE ,TNO en Deltares, 94 pp., 2011.

Vos, P. C., Bunnik, F. P. M., Cohen, K. M., and Cremer, H.: A staged geogenetic approach to underwater archaeological prospection in the Port of Rotterdam (Yangtzehaven, Maasvlakte, The Netherlands): A geological and palaeoenvironmental case study for local mapping of Mesolithic lowland landscapes, Quaternary International, 367, 4–31, https://doi.org/10.1016/j.quaint.2014.11.056, 2015.

Vos, P. C., Van der Meulen, M., Weerts, H., and Bazelmans, J.: Atlas of the Holocene Netherlands, landscape and habitation since the last ice age, Amsterdam University Press, Amsterdam, 96 pp., 2020.

Ward, I., Larcombe, P., and Lillie, M.: The dating of Doggerland–post-glacial geochronology of the southern North Sea, Environmental Archaeology, 11, 207–218, 2006.

Ward, S. L., Neill, S. P., Scourse, J. D., Bradley, S. L., and Uehara, K.: Sensitivity of palaeotidal models of the northwest European shelf seas to glacial isostatic adjustment since the Last Glacial Maximum, Quaternary Science Reviews, 151, 198–211, https://doi.org/10.1016/j.quascirev.2016.08.034, 2016.

Weerts, H. and Berendsen, H.: Late Weichselian and Holocene fluvial palaeogeography of the southern Rhine-Meuse delta (the Netherlands), Geologie En Mijnbouw-Netherlands Journal Of Geosciences, 74, 199–212, 1995.

Wiggers, A. J.: De wording van het Noordoostpoldergebied: een onderzoek naar de physisch-geografische ontwikkeling van een sedimentair gebied, Tjeenk Willink, 1955.

Woldring, H., Boer, P. de, Gillavry, J. N. B.-M., and Cappers, R. T. J.: De palaeoecologie van Duurswold (Gr.): vroeg-Holocene landschapsontwikkeling, Paleo-aktueel, 36–44, 2005.

Wolfert, H. P. and Maas, G. J.: Downstream changes of meandering styles in the lower reaches of the River Vecht, the Netherlands, Netherlands Journal of Geosciences, 86, https://doi.org/10.1017/S0016774600077842, 2007.

Zagwijn, W.: Vegetation, climate and radiocarbon datings in the Netherlands. Part I: Eemian and Early Weichselian, Meded. geol. Sticht., NS, 14, 15, 1961.

Zagwijn, W. H.: The pleistocene of The Netherlands with special reference to glaciation and terrace formation, Quaternary Science Reviews, 5, 341–345, https://doi.org/10.1016/0277-3791(86)90195-2, 1986.

Zagwijn, W. H. and De Jong, J.: RGD Pal. Lab. Rap. 814 & 814a - Pollenanalytisch onderzoek en C14-ouderdomsbepalingen + Aanvullende C14-ouderdomsbepaling aan boring Zwaagdijk Oost 19F/70, Rijks Geologische Dienst, Haarlem, 1982.

Zonneveld, I. S.: De Brabantse Biesbosch: een studie van bodem en vegetatie van een zoetwatergetijdendelta, Wageningen University and Research, 1960.