# Peer review of "HOLSEA-NL: Holocene water level and sea-level indicator dataset for the Netherlands"

_Earth System Science Data, 2024_

## Author Response (AR1)

Dear Editor,

Hereby we present the revised version of the manuscript *"HOLSEA-NL: Holocene water level and sea-level indicator dataset for the Netherlands"*.

We thank the editor and the two reviewers for their useful comments, which have helped to improve the manuscript. In the following sections we have copied the Reviewers comments, followed by our response to those comments including the changes made in the revised manuscript. In addition to changes based on the comments by the reviewers, we have updated some sections of the manuscript to be up to date with the latest version of the HOLSEA-NL dataset, which accompanies this manuscript.

**Review 1 comments**

(1) Can you ignore the compaction of the Pleistocene? It is necessary to explain that compaction can be ignored when it is composed of sand and gravel.

(2) It is easier to understand if you add dotted lines of axis of paleo-valleys in Figure 1a.

(3) Are there no brackish peats?

(4) How were the errors in Table 1 calculated? A simple explanation is needed. Also, I am not sure what the difference is between bog and fen in facies, so it would be good to have a photo of the core. An explanation is needed as to how SLIP was estimated from undifferentiated peat.

(5) Lines 305 to 309. 620+50=670 basal peat data points, which doesn't match 640.

(6) It would be better to show the position of the cross section in Figure 3 in the index map.

(7) Line 398. How were marine carbon and delta R estimated? How many shells did you used for SLIP?

(8) Estimate of compaction is not clear. How was the amount of compaction calculated from the decompaction factor? More detail explanation is required.

**Review 2 comments**

(1) The authors mention including new datapoints from the North Holland Flevoland fringe region but do not include a specific section describing those data. Suggestion to either include a section describing those data or to explicitly describe why that section does not exist.

(2) L520: the authors mention an optional correction related to long-term tectonic and sediment loading related subsidence and cite previous work on this background rate. Given the level of detail of this manuscript, it would be helpful to explicitly state what constraints are used to produce this long term subsidence rate, which must be older than MIS-5e if it was also used for a Last Interglacial sea-level database.

(3) On my first read through, a number of small typos popped out. These errors were small and do not detract from the manuscript's quality. While I do not have time to read the full 35 page manuscript a second time in depth to catch those typos, I encourage the authors to proof their manuscript prior to acceptance.

(4) The article would be improved by citing all of the data sources in the main text. This would give the original papers more credit and reduce the hassle for readers who want to track individual

papers down. The article would be further improved by providing a list of the papers that were excluded, including papers presenting "samples from intercalated peat layers from shallower positions than the basal peats" (L187), "Studies sampling peat at inland locations above +1 m MSL" (L193), and studies presenting samples from pingos (L195). The authors make the good point that this dataset should ideally be a living one with regular updates. Documenting papers that were examined but excluded will make easier the efforts of future workers who want to improve this database.

**Authors' Response to Comments**

**R1C1:**

Line 195, added additional clarification:

"*This Pleistocene surface was exposed before peat formation set on, and had experienced initial compaction and pedogenic consolidation since deposition. Residual compaction of the Pleistocene deposits and those below* (see Kooi et al., 1998) *is considered a part of the separately specified tectono-sedimentary background subsidence component (see Sect. 4.1.3).*"

**R1C2 & R1C6**

Additional visual guides were added to Fig. 1a, as was suggested in these two comments.

**R1C3:**

Line 416, added a paragraph on peats from brackish conditions:

"*Overall, basal peats formed under freshwater conditions. Towards the top of these peat beds, reed-dominated layers often show evidence of brackish storm surges, identifiable by clayey deposits rather than shifts to salt-tolerant vegetation. The mud layers overlying the top of the basal peat usually indicate the change to permanent brackish conditions. Reed is a plant species tolerant to a broader range of conditions and may grow under slightly brackish and mud accumulating conditions (Bos et al., 2012). However, where reed formed basal peats in the transgressive setting of the Netherlands during the Middle Holocene, it is not considered a brackish peat. Bos et al. (2012) found some sites with marine shells in gyttja deposits in the westernmost part of their study area, indicating that some organic layers formed under brackish conditions. During the Middle and Late Holocene, reed stands along river mouths expanding into lagoons may have accelerated the transition from brackish, shallow water to fully terrestrial conditions. This process explains succession patterns within intercalated peat layers, as showcased for the Old Rhine mouth by Pierik et al. (2023), using, amongst others, diatom analysis. Such settings, however, do not apply to the shorter-lived basal peats in our database, in which brackish peats are not distinguished.*"

**R1C4:**

Line 296 onwards, clarified the palaeo-water-depth uncertainty from Table 1 and added examples clarifying the differences between bog and fen peat types and how palaeo-water depth is determined for different peat types, including undifferentiated peats.

"*The palaeo-water depth specification and uncertainty (Table 1) are based on the range of multiannual variation in the seasonally fluctuating water levels. For example, bog peats are ombrogenic, mossy, primarily rain-fed peat bodies, formed around a local water table (palaeo-water depth = 0 ± 0.1 m) perched just above regional water levels. Fen-wood and fen peats are formed in varying hydrological*"

settings: rain, river, and/or seepage-fed. Their palaeo-water depth corresponds to a regional water level, graded to inland past water levels from rivers and seepage zones and to lagoonal and deltaic flood basin water levels in the coastal zones. Fen-wood peats in the Netherlands are typically Alder wood dominated, though they also contain moss, sedges, and reeds, reflecting the vegetation of former swamps, particularly common in river-flooded areas. In these environments, dead plant material accumulated on the peat's surface layer (the acrotelm), where the groundwater table remained at or near the surface for most of the year (palaeo-water depth = 0 m ± 0.1 m). Fen peats are often sedges and reed-dominated, with dead plant material accumulating just underwater and with an estimated acrotelm palaeo-water depth of 0.3 ± 0.2 m. This water depth varies depending on composition and site type, e.g. for "Fen peat on inland dune flanks", palaeo-water depth = 0 ± 0.2 m. For undetermined peat types, an intermediate estimated palaeo-water depth is assumed with a slightly larger uncertainty (0.2 ± 0.3 m)."

**R1C5:**

Line 343 (line 305 in previous version): corrected to ~20 instead of ~50

**R1C7:**

Line 445, added information on the Marine Curve and delta R used:

"Because a regional offset for the reservoir age (delta R) is not established for the study area (W-NW NL), Marine20 was used without an additional delta R, as in Vos & Nieuwhof (2021). We note that this applies to the 12 Late Holocene shell dates regarded as SLIPs from the Wadden Sea (Vos & Nieuwhof, 2021), and to the six LLD shell dates and one tidal ULD from a North Sea incursion into the Overijsselse Vecht palaeovalley, away from direct exchange with Rhine waters (Middle Holocene, Van Straaten (1954)). Including other shell dates from incursions into the Rhine-Meuse palaeovalley was not attempted because of unresolved delta R issues for the Rhine-Meuse mouth."

**R1C8:**

Line 513, added an example of how the decompaction factor is calculated + additional clarification in lines 524 and 525:

"We present an example from the first category to demonstrate how the decompaction factor affects the compaction correction, expressed as an offset to sample depth. A decompaction factor of 3 implies that, at the time of inundation, the peaty layer between the sample and the Pleistocene substrate was three times thicker than its current thickness (T), recorded as "Depth-to-consolidated-surface" in Field 18. Therefore, the upward offset stored in Field 64 should be 3 times the current elevation above the consolidated surface, which equals 2 times the thickness from Field 18 (T + 2T = 3T; thus, 3T − T = 2T). For decompaction factors of 2 and 2.5, the multiplier used in Field 64 is 1 and 1.5, respectively."

**R2C1:**

The new data points from the North Holland Flevoland fringe region are described in Sect. 5.2 paragraph 4. We have not adjusted this section, but we have added some references to this section and to the HOLSEA-NL dataset (e.g. Line 704), to emphasize the newly added data.

R2C2:

Line 595, added more information on the input used for the long-term tectonic and sediment loading related subsidence correction (see text below). Furthermore, we have added additional emphasis in other parts of the manuscript (mainly Sect.5 & 6) to clarify which version of the data was used, with or without the tectonic correction.

*"As input to this correction, a map product specifying a background rate was used, following the approach of Cohen et al. (2022: their Sect. 3.3 Vertical Land Motion) in their Last Interglacial sea-level database. This map considers estimates of long-term mean subsidence rates calculated over the 1.8 Myr, derived from onshore and offshore Quaternary basin fill mapping, along with an associated uncertainty specification. For more details see Cohen et al. (2022) including their Sect. 6.6 on preferring 1.8-Myr averaged rates, which are 80-70% that of 2.6-Myr rates. The spatial patterns and values are consistent with earlier tectono-sedimentary back-stripping analyses for this region (Kooi et al., 1998; producing rates calculated over the last 2.6-Myr) and applications thereof in relative sea-level data analysis in Kiden et al. (2002), Vink et al. (2007) and Simon and Riva (2020)."*

R2C3:

Several corrections of typos, better interpunction, consistency of tense etc. were made (see track-changes document).

R2C4:

We have taken up the first recommendation to implement more citation of data sources in the main text (also a policy of the ESSD journal), and this has included citation-pointers to work adjacent, but outside the scope of our data base activity in Sect. 3.1. Providing a full list of papers known to us and holding such data we did not do, as the wealth of potential additional data that exists makes it nearly impossible to report all examined but excluded data. In the revised submission, we have included a new table (Table 2) that lists all the primary data source references, in sync with the HOLSEA-NL dataset, and placed this at the end of the manuscript.

Regarding the shared author and reviewer point that this dataset should ideally be a living one with regular updates, we like to stress that, ideally, the HOLSEA-NL database indeed holds any data points with meaning to past water levels (groundwater table, sea-level), but that it is not the format to store radiocarbon dates of e.g. lake fills, tidal incursion events, river flooding events, river branch abandonment, inland bog peat history etc. Samples with such meanings need to be administered separately, the HOLSEA template is not of use, and more generic approaches to databasing such information are recommended. The HOLSEA-NL database does contains unique identifier fields (such as sample lab-number) and geographical (x, y, z) and bibliographical fields (citations) that allows to link and cross-verify it to other databases and literature.